# Can Supramolecular Polymers Become Another Material Choice for Polymer Flooding to Enhance Oil Recovery?

**DOI:** 10.3390/polym14204405

**Published:** 2022-10-18

**Authors:** Linghui Sun, Zhirong Zhang, Kaiqi Leng, Bowen Li, Chun Feng, Xu Huo

**Affiliations:** 1Development Research Institute, Research Center for Enhanced Oil Recovery of China Petroleum Exploration, Beijing 100089, China; 2Institute of Seepage Fluid Mechanics, Chinese Academy of Sciences, Langfang 065000, China

**Keywords:** supramolecular polymer, enhanced oil recovery, anti-temperature and anti-salinity, stimulus response, viscoelasticity

## Abstract

High molecular polymers have been widely studied and applied in the field of enhanced oil recovery (EOR). At present, the focus of research has been changed to the design of polymer networks with unique properties such as anti-temperature and anti-salinity, good injection and so on. Supramolecular polymers have high viscoelasticity as well as excellent temperature, salt resistance and injection properties. Can supramolecular polymers become another material choice for polymer flooding to enhance oil recovery? The present review aims to systematically introduce supramolecular polymers, including its design strategy, interactions and rheological properties, and address three main concerns: (1) Why choose supramolecular polymers? (2) How do we synthesize and characterize supramolecular polymers in the field of oilfield chemistry? (3) What has been the application progress of supramolecular polymers in improving oil recovery? The introduction of a supramolecular interaction system provides a new idea for polymer flooding and opens up a new research direction to improve oil recovery. Aiming at the “reversible dynamic” supramolecular polymers, the supramolecular polymers are compared with the conventional covalent macromolecular polymer networks, and the challenges and future research directions of supramolecular polymers in EOR are discussed. Finally, the author’s viewpoints and perspectives in this emerging field are discussed.

## 1. Introduction

Enhanced oil recovery (EOR) is a general term for oil recovery by improving the physical and chemical properties of reservoir and fluid, improving swept volume and oil displacement efficiency [1,2,3,4]. With the continuous consumption of oil and the increasing demand for energy, it is particularly important to improve the crude oil recovery of difficult to recover reservoirs such as mature oilfields and low permeability reservoirs [2]. After primary and secondary exploitation, more than 50% of the crude oil in the reservoir still needs to be exploited by effective methods. At present, most oilfields have entered the stage of tertiary oil recovery; the enhanced oil recovery of tertiary oil recovery is mainly aimed at the remaining oil and residual oil in the formation after secondary oil recovery, in which chemical-enhanced oil recovery (CEOR) is one of the important means to improve oil recovery, including polymer flooding, surfactant flooding, nanofluid flooding and composite flooding [5,6,7,8].

Polymer flooding accounts for more than 77% of global CEOR projects, and it is the main technology to improve oil recovery in the middle and later stage of mature oilfield development [1,3]. It is generally believed that polymer flooding can improve oil displacement efficiency mainly by reducing the water–oil mobility ratio and expanding sweep volume. Although a comprehensive and perfect mechanism of polymer flooding has not been given at present, through a large number of literature and field experiments [5,9,10,11,12,13], we can draw a basic conclusion: oil recovery depends to a large extent on the high viscoelasticity of polymers. With the deepening exploitation of oil fields, the complexity of the reservoir increases, the reservoir environment becomes more and more harsh, and the conventional polyacrylamide for oil displacement is faced with some problems, such as weak tackifying ability, easy degradation under high temperature, high salinity and so on. In the 1980s, Evani and Roselti proposed the concept of hydrophobically associating polymers for the first time [14]. After that, scientists around the world have conducted sufficient research on acrylamide-based modified polymers. By increasing the molecular weight, introducing temperature-resistant and salt-resistant groups such as hydrophobic monomers, anions and cationic groups, the shielding effect of high valent metal ions such as Ca^2+^ and Mg^2+^ on the molecular chain of polyacrylamide was effectively reduced [15,16], and the viscoelasticity of the polymer at high temperature was improved. The chemically modified temperature-resistant and salt-resistant polymers bring great flexibility to improve oil recovery and change the restrictions on the use of polymers in high-temperature and high-mineralization reservoirs. and it shows great application potential in drilling construction, hydraulic fracturing, profile control, wastewater treatment and other fields [17,18,19,20].

For most of these areas, the fast advancements are largely driven by the increasing synergy between the development of basic materials and the expansion of oil field chemical engineering. For example, the discipline of polymers and enhanced oil recovery are cross-integrated and promoted each other. On the one hand, the development of polymer materials can broaden the prospect in the field of enhanced oil recovery. On the other hand, the significant improvement of crude oil recovery in turn brings more possibilities for polymer network design. However, the polymer network of existing enhanced oil recovery is mainly formed by covalent bond cross-linking, and the static and permanent polymer network structure is irreversible in the face of shear, high temperature, high salinity and so on. No more crude oil will be produced [2,9,21]. Although scientists can enhance the resistance of polymer after chemical modification, they still have to face the contradiction between polymer flooding ability and injectivity: it is necessary to synthesize low molecular weight polymer while maintaining high viscosity of polymer solution. On the premise that the polymer solution can be injected, we do not want the polymer chain degradation to cause viscosity loss.

In the past few decades, non-covalent supramolecular systems have been introduced as an alternative to conventional covalent polymers, and their high viscoelasticity, temperature and salt tolerance and multi-functional stimulus response have shown good applicability in reservoirs. Supramolecular polymer comes from monomers forming highly ordered polymer arrays driven by dynamic and reversible non-covalent forces (hydrogen bonding, host–guest interaction, π–π stacking, etc.) [22,23]. Polymer chains will be entangled with each other and maintain high viscoelasticity, showing similar rheological properties to covalent polymers [24,25,26]. The reversible dynamic characteristics of supramolecular polymers have played a key role in life sciences, pharmaceutical engineering, energy science and so on [22,27,28,29]. At present, various supramolecular polymers have been reported and used in oil field chemical engineering as an important supplement to conventional polymers [30,31,32]. More and more researchers with different research backgrounds introduce non-covalent interactions into polymer networks, which are used in some reservoirs where conventional polymers are difficult to apply.

Considering that supramolecular polymers have both the mechanical and rheological properties of polymers and the dynamic reversibility and stimulus response of supramolecular systems, when supramolecular chemistry is combined with enhanced oil recovery engineering, its potential seems infinite. It can be predicted that supramolecular polymer flooding will continue to promote the development of many disciplines in the future, especially the enhanced oil recovery of low-grade reservoirs. Since the beginning of this century, scientists have synthesized a series of supramolecular systems based on non-covalent interaction [32,33,34], which are used to improve the tackifying ability and temperature and salt tolerance of polymers. The combination of supramolecular chemistry and enhanced oil recovery engineering has led to their synergetic, coordinated development.

There has been some work on the application of supramolecular polymers in improving oil recovery. First, some scientists modified polyacrylamide, which can be divided into the following categories: cyclodextrin-based polymers [30,35,36,37], amphiphilic polymers [38,39], hydrophobically associating polymers [40,41], multicomponent composite polymers [36,42], temperature-resistant and salt-tolerant monomer polymers and so on. For example, Xu et al. [43] developed a water-soluble cationic copper phthalocyanine supramolecular oil displacement agent WMM-100. This supramolecular oil displacement agent mainly forms a supramolecular conjugate system through the interaction between cationic groups with large radius and negatively charged rock surface and crude oil (hydrogen bond, stacking effect, coordination bond, electrostatic interaction, etc.). In order to improve the effect of microscopic oil displacement, some results have been achieved in the experiment of enhancing oil recovery in the laboratory. In recent years, the host–guest system has been widely studied and used to construct a variety of oil displacement systems. Zou et al. [44] synthesized a cyclodextrin-based polymer and proposed a mixed flooding of host–guest system and surfactant, which not only reduces the interfacial tension of rock–oil–water but also fully controls the oil–water mobility ratio and improves the oil displacement efficiency of mixed flooding. A large number of studies [39,45,46] have shown that the modified polymers have specific application requirements in terms of structure and properties, and they have different effects in improving oil recovery. The study of supramolecular chemical systems in petroleum has played an important role in promoting the supramolecular system in improving oil recovery. However, these studies mainly focus on the medium and high permeability reservoir system, and they study the polymer network with temperature and salt tolerance and a substantial increase in viscosity.

Specifically, we focus on another aspect of the combination of supramolecular polymers and enhanced oil recovery: that is, the application of supramolecular polymers in low permeability reservoir systems, including the design and synthesis of supramolecular polymer networks suitable for low permeability reservoirs, the percolation law and characterization methods of supramolecular systems in porous media, and the specific performance of stimulus-responsive polymers in harsh reservoir environments. Based on this, we are actively exploring the use of supramolecular polymers to completely replace conventional polymers in enhanced oil recovery and make them used in low permeability reservoirs or even use matrix crude oil. Is it feasible to introduce the polymer of supramolecular interaction system as the basic material to improve the oil recovery of low-grade reservoir oil? At present, there is a lack of systematic establishment of the concept and corresponding research direction of a supramolecular polymer–EOR system, and there are few studies on the design of supramolecular polymers and the self-assembly structural mechanism in porous media in oilfield chemistry. In most of the work, the research on enhanced oil recovery based on supramolecular chemistry is mainly focused on improving the temperature and salt tolerance and low concentration tackifying ability of polymers. However, there is little research on the flexible application of stimulation response, self-assembly and self-healing properties of polymers in harsh reservoir environments.

In this review, we envision that the supramolecular system, as the main material for enhancing oil recovery, forms a three-dimensional network framework by forming non-covalent cross-linking between polymer chains without any other chemical cross-linking agents, and then, it tries to establish the corresponding characterization methods and application directions in order to provide a comprehensive reference for the development of supramolecular polymer–EOR, especially for low permeability reservoir systems. Here, we define supramolecular polymer–EOR as enhanced oil recovery research only using supramolecular polymer materials, and we temporarily ignore some composite oil displacement systems, including host–guest polymer/surfactant composite flooding, supramolecular nanocomposites and so on. Then, the related problems that supramolecular polymers may face as a new generation of thematic materials in enhanced oil recovery are determined, including why non-covalent interaction is chosen to construct polymer networks, supramolecular monomer screening and low-cost synthesis, characterization, and the application progress of supramolecular polymers in enhanced oil recovery. Although supramolecular polymer enhanced oil recovery is proposed as a relatively new concept, it has been systematically studied in biomedical, material engineering and other mature fields, and the design ideas of supramolecular polymer in other research fields are discussed, providing inspiration for the characterization strategy for laboratory research on enhanced oil recovery.

Efforts are categorized and discussed in the context of supramolecular polymers for enhancing oil recovery from three aspects, as follows: the first part of this review explains the reasons for the study of supramolecular chemistry in the field of enhanced oil recovery. The limitations of conventional covalent polymers (partially hydrolyzed polyacrylamide, xanthan gum, modified polymers, etc.) in material process and field application (charge shielding, polymer chain crimping, high salt precipitation, irreversible degradation at high temperature and shear resistance of flexible chain) were reviewed. The advantages of supramolecular polymers (rich monomer source, high biocompatibility, flexible and rigid polymer chain, stimulation response, reversible dynamic network, excellent rheological properties and good oil displacement performance) are discussed. The second part of this review summarizes the methods of constructing oil field chemical materials based on supramolecular polymers (transplanting the conventional design ideas of supramolecular research field to oil field chemical materials, supramolecular modification of conventional oil displacement polymers, and developing characterization strategies for supramolecular oil field chemical materials). This paper focuses on the feasibility and limitations of the host–guest system, hydrophobic polymers and ionic amphiphilic polymers in improving oil recovery in recent years. On this basis, improvements to the existing conventional polymers are proposed and designed to improve the applicability to harsh reservoir environments. This paper also reviews the conventional characterization methods of oil field chemical materials in oil field chemical engineering, and it puts forward the application of supramolecular self-assembly mechanism research technology in oil field chemistry. At this time, supramolecular polymers have a wider research direction in structure, function, characterization and application. In the third part of this review, the applications of supramolecular polymers in enhanced oil recovery are enumerated and prospected. Compared with covalent polymers, the incompatibility between supramolecular polymers and oilfield chemistry is emphasized. Finally, we share our conclusions and our perspectives for the future.

## 2. Why Choosing Supramolecular Polymer as the New Material for Enhanced Oil Recovery?

### 2.1. Conventional Polymers Utilized for EOR

Polymers used in EOR mainly include natural polymers and synthetic polymers. Among the synthetic polymers, the most representative is polyacrylamide, which is widely used in the fields of oil exploitation, wastewater treatment, biomedicine, chip manufacturing and so on [3,47,48]. Polyacrylamide is a general term of acrylamide and its derived homopolymers and copolymers. As a linear water-soluble polymer, PAM aqueous solution has a certain viscosity. In the practical application of tertiary oil recovery, polyacrylamide is generally partially hydrolyzed (the degree of hydrolysis is usually 25–35%) to obtain more excellent performance [49]. It is found that the introduction of hydrolysis groups makes the polymer transform into polyelectrolyte, which is helpful to the extension of the molecular chain, and at the same time, it avoids a large amount of adsorption on the surface in the formation pores. The rheological properties of HPAM are greatly affected by the formation environment, so other synthetic polymers have also been applied to enhance oil recovery, including branched poly (propylene amide) [50], N, N-dimethylacrylamide copolymer [51], and HPAM derivatives with temperature and salt-resistant monomers (such as AMPS and NVP) [52,53,54]. No matter whether HPAM or other synthetic polymers, the polymer molecular chain has the characteristics of flexibility, easy change of molecular conformation, and extremely easy to bend and entangle. It can build a rich network structure such as honeycomb, ribbon, dendritic and network, so it has high viscoelasticity and good rheological regulation ability.

Natural high molecular polymer, also known as biopolymer, is a polymer produced by microorganisms, plants and other biological products. As the most widely used biopolymer in the petroleum industry, xanthan gum has been proposed for EOR as early as the end of the 1950s [55,56,57]. Ghoumrascibarr et al. [58] proposed that xanthan gum is mainly a pentasaccharide repeat unit composed of d-mannose, d-corn glucose, and d-corn glucose furfural. The good salt resistance and stability of xanthan gum are attributed to the formation of dimer and double helix structure. Under the condition of high salinity, the aqueous solution of high concentration cation is strongly polar, which indirectly enhances the nonpolar effect between xanthan gum molecules and keeps the viscosity of xanthan gum stable. At the same time, the cation generally interacts with the negatively charged glucuronic acid group on the side chain of xanthan gum, reducing the influence on the polymer main chain. Under acidic conditions, pyruvate and acetyl groups of xanthan gum begin to be removed, which is conducive to the formation of a network structure, and the phase transition temperature of the solution increases; Under alkaline conditions, the acetyl group of xanthan gum begins to be removed, which will hinder the formation of a network structure. The phase transition temperature of the solution becomes smaller, but the effect on the viscosity of the solution is small [59]. In addition, other biopolymers have gradually been used in the petroleum industry, including guar gum, Weilan gum, hard core dextran, cellulose, pleated polysaccharide, lignin and mushroom polysaccharide [3,57,60,61]. Because of its stability in extreme environments, it has great potential to construct a supramolecular self-assembly system with biopolymer as the main chain.

### 2.2. Limitations of Traditional Polymers in EOR

With the deepening of reservoir exploitation and long-term strong injection and production, the focus of exploration and development has been shifted to low-grade reservoirs with poor crude oil quality, complex storage conditions and complex distribution characteristics, including low permeability reservoirs, heterogeneous reservoirs, heavy oil (ultra-heavy oil), marginal small oil and gas fields and high water cut reservoirs in old oil and gas fields [21,62,63]. However, there is a gap between the development of oil displacement polymers and the deterioration of reservoir conditions. The existing polymer materials show poor temperature and salt resistance and shear resistance in extremely harsh reservoirs [9,64]. Due to the irreversible degradation of the static permanent covalent polymer network, it is far from meeting the viscosity of the displaced crude oil.

At the same time, the injectability of polymer is the key to determining whether it can carry out chemical flooding. Cheng et al. [65] studied the matching relationship between the size of the random coil in polymer solution and the reservoir pore throat:r_H_ = [K(1 − *Φ*)^2^/C]^0.5^,R_G_ = 0.619{M × [ƞ])/10^24^}^1/3^r_H_/R_G_ > 5(1)

If permeability K = 200 × 10^−3^ μm^2^, rH = 1.52 μm. In order to maintain the injectability of the polymer, we must ensure that the molecular weight is not more than 10 million, which limits the viscosity-increasing ability of the polymer. Therefore, for the conventional covalent polymer, the contradiction between high viscosity and injectability limits its application in low-permeability reservoirs [13,66,67]. Recently, people have focused on stimulation-responsive materials in material engineering and biomedicine for EOR research, and they actively explored dynamic polymer networks with good oil displacement performance to build a new generation of oilfield chemical materials.

### 2.3. Supramolecular Polymer: Infinite Network based on Non-Covalent Action

The concept of the supramolecule began with the research work of Professor Lehn in 1990 [68]. There are many research studies in the field of life and material science, and the development is rapid. With the promotion of science and technology, the trend of supramolecular chemistry and other disciplines to solve more practical problems is also growing. Supramolecular polymer science, which is the combination of supramolecular chemistry and polymer chemistry, is one of them. The supramolecular polymer is a complex ordered, organized and highly oriented chemical system formed by non-covalent bonds between molecules. Driven by non-covalent interactions, supramolecular polymers are cross-linked and entangled to form an amorphous network. Their physical and chemical properties are between those of conventional supramolecular systems and high molecular polymers [28,69]. Supramolecular polymers not only have the mechanical and rheological properties of high molecular polymers [26], but they also show the reversibility and stimulus response of supramolecular systems in some cases [69,70], which has gradually attracted more and more attention. Supramolecular polymer materials are widely available and relatively cheap. Due to the specific recognition and directionality of non-covalent bonds, scientists can accurately design the topological structure of supramolecular polymers to meet different needs. At the same time, the low cost and biodegradability of supramolecular polymer materials bring the possibility and convenience of easy disposal after use.

With its unique properties, the application of this “reversible dynamic” polymer material in different research fields has increased significantly. In recent years, more and more oilfield chemical materials based on supramolecular polymer networks have been reported in the field of oilfield chemistry, such as deep profile control, drilling construction and oilfield wastewater treatment, which have good performance in viscosity-increasing ability and stimulation response. For example, the high-performance drilling fluid developed by Guancheng et al. [71] of China University, which is made of petroleum and is based on supramolecular chemistry theory, has strong sand-carrying capacity, low filtration loss and excellent temperature and salt resistance. Guancheng et al. developed a supramolecular drilling fluid system with good shear recovery characteristics with a self-made supramolecular cutting agent and supramolecular fluid loss prevention agent as the core. When the bentonite slurry: supramolecular fluid loss reducer: supramolecular cutting agent: plugging agent: NaCl = 1:1.15:1:1.5:15, the supramolecular drilling fluid has good “self-assembly” ability, and the elastic modulus can quickly recover and tend to be stable after high strain. The stimulus responsiveness of supramolecular systems is an excellent proof of the reversibility of non-covalent interactions, which enables supramolecular polymers to respond to various physical, chemical and biological factors, such as light, pH, temperature, ion concentration, etc. Fenpu et al. [72] successfully synthesized a version of amphiphilic hyperbranched polymers (amhpms) by regulating the terminal functional groups of functional monomers with acrylamide and using sodium acrylate as the hyperbranched polymer backbone. It was found that amhpms exhibited rich viscosity response behaviors to external environmental stimuli (shear, temperature, salt, pH). This is attributed to the change of the charge property of the amphoteric branched quaternary ammonium group and the association behavior of the hydrophobic chain, while the hyperbranched structure brings a wider response region for amhpms.

When the conventional covalent polymer is faced with the complex environment of tensile shear, high temperature and high salinity, irreversible degradation will occur [32]. Supramolecular polymers can form complex, ordered and organized dynamic networks by the non-covalent interaction of small molecules, oligomers and polymers. Driven by non-covalent action, monomer units self-assemble into a somewhat ordered topological structure on the time scale of a few minutes to days, and then, they form a solid and dense three-dimensional network structure in non-covalent action, covalent action or entanglement. The supramolecular aggregation behavior and molecular rigidity of many functional monomers can inhibit the crimp degradation of polymer chains under high temperature and high salinity [30,32], and their reversibility enables the polymer network to undergo dynamic reorganization after shear failure. Therefore, the supramolecular oil displacement system shows good applicability in low-grade reservoirs. In the past decade, different polymer networks driven by non-covalent interaction have been used to enhance oil recovery, which preliminarily proved the feasibility of supramolecular polymers as a new generation of host materials for enhanced oil recovery polymer flooding.

### 2.4. Differences between “Supramolecular Weight Polymers” and High Molecular Weight Polymers

Supramolecules are defined as molecular aggregates formed by two or more kinds of molecules through non-covalent interaction, which is superior to covalent molecular aggregates in performance, such as molecular recognition, reversible dynamics, stimulus response, etc. Therefore, in the supramolecular chemistry field, the concept of a “supramolecular weight polymer” is not widely mentioned. At the same time, in the early stage of polymerflooding research, in order to improve the viscoelasticity of oil displacement polymers, scientists generally synthesize ultra-high molecular weight polymers with high viscoelasticity by increasing the molecular weight. However, some researchers have ambiguous understanding of supramolecular polymers due to their academic background, and they have confused supramolecular polymers with ultra-high molecular weight polymers. From the perspective of definition, scientists believe that supramolecules are based on intermolecular forces rather than on “degree of polymerization” and “molecular weight” like polymers.

It seems that we can divide supramolecular polymers into low molecular weight supramolecular polymers, high molecular weight supramolecular polymers and ultra-high molecular weight supramolecular polymers, as shown in Figure 1. Driven by non-covalent interaction, there is physical cross-linking between low molecular weight polymer chains, and the resulting molecular aggregates have an ideal three-dimensional topological structure, showing the mechanical and rheological properties of polymers.

### 2.5. Major Superiority of Supramolecular Polymer Flooding in Enhanced Oil Recovery

Supramolecular polymers suitable for oilfield chemistry belong to random coil polymers. Their properties and rheological properties are similar to those of conventional polymers, but they have supramolecular assembly dynamics. Figure 2 visually shows the main advantages of supramolecular polymers in EOR and the main difficulties faced by traditional polymers.

First, driven by non-covalent interactions, different monomers or polymer precursors associate with each other, providing a large number of physical cross-linking points for the construction of polymer networks. Compared with linear polymers, the polymer chains in supramolecular systems are cross-linked and entangled to form a more solid and dense complex network structure. Therefore, the supramolecular polymer can exhibit excellent viscosity-increasing ability with low relative molecular weight.

Second, supramolecular polymers can handle excessive heat energy, improve the applicability in high-temperature reservoirs, and show better thermal stability than HPAM. On the one hand, the rigid side groups of supramolecules are distributed along the polymer backbone. Compared with the flexible polymer backbone, the rigid chain can keep the molecular conformation unchanged at high temperatures and reduce the viscosity loss. On the other hand, due to the introduction of a large number of physical cross-linking points into the polymer network, supramolecular units will gather under the drive of non-covalent interaction (such as host–guest interaction, hydrophobic association, which are endothermic) when the temperature rises and maintain the three-dimensional topological structure.

Third, reversible dynamic properties are the key characteristics of supramolecular polymers. In the non-covalent system, the formation and destruction of physical cross-links in polymers can be bidirectional. Compared with covalent polymers based on chemical cross-linking, supramolecular polymers can self-assemble under certain conditions after the polymer network is damaged by environmental factors (high temperature, high salt, extreme pH, tensile shear) in the reservoir, and the polymer network is rearranged and formed so as to restore viscoelasticity and displace more crude oil.

Fourth, stimulus responsiveness is one of the most attractive properties of supramolecular polymers. By introducing responsive groups such as temperature and pH, researchers can control the self-assembly process of polymers and express specific response behaviors at the macro level. Up to now, different stimulation response systems have been gradually developed and applied in EOR research, including the salt viscosity-increasing system, temperature-adaptive polymer, CO_2_-responsive gel, etc. Stimulation-responsive supramolecular polymers are expected to achieve the “targeting” effect on specific reservoirs and achieve the goal of efficient production.

Fifth, the monomers or precursors used to construct supramolecular self-assembly structures have a wide range of sources, low cost, and most of them are naturally degradable and environmentally friendly. When the supramolecular polymer is injected into the formation, it will increase the viscosity or gel without a chemical cross-linking agent, which avoids the chromatographic separation of multi-chemical components in porous media. However, for traditional oil displacement polymers, most petroleum-based polymer chains are not rapidly degradable. At the same time, the use of a chemical cross-linking agent may damage the reservoir and cause environmental pollution.

Meanwhile, the combination of abundant multi-level cross-linking networks and reversible dynamic characteristics in supramolecular polymers provides support for the good injectability of high viscosity polymer solution in low-permeability reservoirs. Yan et al. [73] put forward the self-adaptive concept for the molecular design of polymer-enhanced oil recovery in low-permeability reservoirs, and they revealed the self-assembly process of a supramolecular network based on hydrophobic interaction through porous media, as shown in Figure 3. The hydrophobic interaction between the side chains is destroyed when the polymer thread group is subjected to shearing and stretching through the pore throat. Due to the introduction of rigid ionic groups, the conformation of the polymer chain can be maintained through the pore throat. Before flowing through the next pore throat, the supramolecular aggregates can re-form and restore the viscosity, realizing the dissociation re-aggregation process of the polymer through the pore throat, thus avoiding the viscosity loss during injection and flow in the porous medium. The complex and changeable reservoir environment and oil field construction site make the conventional covalent polymers face great challenges. The multi-cross-linking strategy and versatility of supramolecular polymers support the research idea of building various polymer networks. The introduction of supramolecular polymers provides strong technical support for the development of various complex reservoirs. The author gives a comprehensive and in-depth introduction to the molecular structure design, synthesis, characterization and performance of supramolecular polymers under various reservoir conditions, which is of great significance to the large-scale research of supramolecular polymers in oil fields.

## 3. How to Fabricate Supramolecular Polymers?

In the research of EOR, the design and synthesis of the polymer is the first key step for various reservoir applications. It will be a great challenge to use the technology and tools of oilfield chemical engineering to achieve the complexity and adaptability of supramolecular polymers in traditional application fields. In this review, we focus on the network design, monomer selection and synthesis methods of supramolecular polymers, introduce the characterization methods of supramolecular polymers as oilfield chemical materials, and propose improvements and suggestions to enhance the compatibility of supramolecular polymers with EOR applications.

### 3.1. Synthesis of Supramolecular Polymers Suitable for EOR

The key to preparing supramolecular polymers is to select a suitable main chain and potential side chain structure, and introduce non-covalent interactions that can form dynamic cross-linking points. We must consider the characteristics of non-covalent type, polymerization activity of monomers and economic cost in great detail, because the supramolecular network structure and functional regulation of polymers are highly dependent on these elements. Especially in the application of EOR, the dissociation rate of supramolecular bond and reservoir applicability may be of great importance. Through intelligent screening of monomers and construction of supramolecular modules, oilfield chemical materials with required structure and function can be obtained.

#### 3.1.1. Types of Non-Covalent Interactions for the Construction of Supramolecular Polymers

The driving force between supramolecular polymers involves a variety of non-covalent interactions, including hydrogen bonding, host–guest interaction, metal complexation, hydrophobic interaction, electrostatic interaction, and π–π stacking [22,23]. The strength of different non-covalent interactions varies greatly, so we can choose different functionalized non-covalent interactions to cross-link polymers and then adjust the network structure of supramolecular polymers to obtain materials with different mechanical properties. Here, we only give a very brief introduction to each common non-covalent effect used for supramolecular polymers, including the advantages and limitations in EOR applications.

The hydrogen bond is a kind of dipole attraction between the hydrogen atom next to the highly electronegative atom and the adjacent molecule. It plays an important role in many biological processes, such as DNA base pairing, enzyme catalysis and protein folding [70,74,75]. It has become the most widely used non-covalent interaction in the synthesis of supramolecular polymers and reversible cross-linked polymer networks. The hydrogen bond is a unique non-covalent bonding phenomenon, which has directionality, saturation and versatility [76]. In the bonding process of intermolecular hydrogen bonds, hydrogen bonds can guide and regulate the structure of molecular aggregates, change the degree of freedom of supramolecular polymer chains through the number of hydrogen bonds, and change the polymer network structure [77,78]. In addition, the directionality of the hydrogen bond also endows the recognition process between molecules with high selectivity and specificity. Although in the water environment, polar water molecules compete with hydrogen bond complementary units for binding sites to weaken the hydrogen bond strength; at present, researchers have improved the bonding strength by building multiple hydrogen bond arrays and building hydrophobic micro domains to isolate water molecules [79,80,81,82].

π–π stacking is common among aromatic organic compounds, which is caused by the overlap of p orbitals in π-conjugated systems. π–π stacking interactions are generally provided by aromatic systems, while flexible alkyl side chains are helpful to improve the solubility of monomers and polymers, and they sometimes can provide the solvation and hydrogen bonding of supramolecular polymers [79]. The supramolecular system based on π–π stacking interaction is generally composed of a dished core composed of a planar aromatic system and a side chain composed of a flexible alkyl chain. The delocalized π electron coupling associated with π–π stacking may cause optical and electronic properties such as semiconductivity and exciton transport. Therefore, the materials constructed based on this interaction are often used in the fields of optics and biomaterials [80,82].

The metal coordination bond is mainly reflected in the force between the metal and the ligand. The strength of the force is similar to that of the covalent bond, and it has high directivity and stability [83]. Supramolecular polymers can form a stable three-dimensional network structure through the complexation of side chain groups and high valent metal ions [28,83,84,85], which affects its hydrodynamic volume and plays a role in viscosity enhancement. Santo studied the transition of polymer viscoelastic fluid sol–gel through molecular dynamics simulation and proved that the complexation between metal ions and ligands can form physical cross-links and change the rheological properties of polymers.

In 1959, Walter pointed out for the first time that hydrophobic units in aqueous solution have the effect of spontaneous aggregation hydrophobic interaction. As a trend of oil and water separation, it leads to the spontaneous assembly of hydrophobic groups and hydrophilic groups located in aqueous solutions. For example, surfactants in aqueous solutions self-assemble to form micelles of different shapes driven by hydrophobic interactions, in which the interior of micelles is stabilized by hydrophobic forces [86]. In the 1980s, Evani and Rose first introduced hydrophobic groups into polymers, giving polymer solutions special properties. Subsequently, in terms of EOR, the outbreak of the oil crisis promoted the research of hydrophobic association polymers, mainly including hydrophobic-modified polyacrylamide, hydrophobic-modified cellulose and hydrophobic-modified alkali expanded lotion [73,86,87]. At the same time, a hydrophobic polymer surfactant binary composite system was developed to further improve the viscosity of polymer through surfactant [88].

Electrostatic interaction is a non-covalent bond between charged particles. Its strength can be as high as 350 kJ/mol, which is closer to the strength of the covalent bond [32,89]. However, electrostatic interaction generally does not use directivity and saturation. Electrostatic interaction is often used to construct zwitterionic polymers. By introducing anionic groups and cationic groups into polymer chains, zwitterionic polymers can exhibit a salt viscosity-increasing effect in high salt environments [90]. In addition, researchers often use electrostatic interaction as the driving force to construct self-assembled supramolecular systems responsive to pH and CO_2_ stimulation [91,92].

Host–guest interaction is the driving force for the complexation between host and guest molecules, which usually includes the superposition of the above-mentioned non-covalent interactions [93,94]. Compared with other non-covalent interactions, the host–guest interaction has a high degree of molecular structure matching, which requires the size and shape of the host cavity structure and the mutual matching between the host and the guest, and it becomes a supramolecular interaction with high recognition ability. The cyclodextrin-based supramolecular system is widely used in EOR [30,34,37,95,96,97]. When the host–guest interaction is used to construct the supramolecular polymer system, the enhanced polymer network structure makes the hydrodynamic volume of the solution increase dramatically. At the same time, because the cavity of cyclodextrin has strong binding force on the hydrophobic guest molecules with appropriate molecular size, its viscosity-increasing effect is better than that of the hydrophobic association polymer. Under the conditions of high temperature and high salinity, the host–guest system improves the sensitivity of polymer chains to high temperature and high salt, and in some cases, it also exhibits salt sensitive viscosity increasing and other stimulus response behaviors.

In addition to the single non-covalent interaction, some researchers have also proposed using multiple non-covalent interactions to prepare supramolecular polymers [32], combining the advantages of each force to avoid the limitations of single supramolecular interaction in structure, performance and stability. These include some recently developed synergistic types, such as hydrophobic interaction-enhancing hydrogen bond [79], multiple host–guest interaction synergistic driving [93], metal coordination hydrogen bond-driven multiple stimulus response systems [98], etc. The development of multiple interactions has greatly enriched the types of supramolecular polymer networks and brought new characteristics to their applications in related fields.

#### 3.1.2. Selection of Precursors for Supramolecular Polymers

Many different polymer precursors can be used to prepare supramolecular polymer networks, including natural polymers, synthetic polymers and the mixed networks [99,100]. Polymer precursors determine the basic physical and chemical properties of supramolecular polymer networks. After chemical modification, polymer precursors can be further used in related fields. For EOR, a popular class of synthetic precursor polymers are water-soluble polymers, including polyacrylamide (HPAM), polyethylene glycol (PEG), and sodium polyacrylate (PAAS). However, many supramolecular polymer networks are unstable in water because many supramolecular modules cannot form strong enough binding motifs to resist competitive hydrogen bonds, and the molecular rigidity of flexible polymer chains cannot cope with shear stretching in the formation. Therefore, natural polymers are often used instead of synthetic precursors to construct supramolecular polymer networks, such as biological polysaccharides based on xanthan gum, cellulose, chitosan and so on [101,102]. These materials are low-cost, environmentally friendly, have excellent salt resistance, and have abundant chemical modification sites. They are expected to be ideal candidates for building supramolecular polymer networks. Of course, natural polymers also have certain limitations: because natural polymers are extracted from plants, microorganisms and other living bodies, with wide molecular weight distribution and different batch compositions, it is impossible to customize natural polymer precursors according to needs. Secondly, natural polymers are easily decomposed by microorganisms in the formation, which limits large-scale application. Therefore, the mixed precursor formed by the combination of synthetic polymer, and natural polymer is usually an excellent compromise.

#### 3.1.3. Design Strategy of Supramolecular Polymer Network

Supramolecular interaction is the most important element in the construction of supramolecular networks. In order to obtain the required materials, we must consider the physical cross-linking motifs and different polymer architectures at the same time. Supramolecular polymer networks combine the characteristics of physical and chemical networks, and they can be customized by flexibly using polymer building blocks and introducing supramolecular motifs, thus forming various types of polymer structures.

Supramolecular association motifs can be divided into homo complementary motifs that are associated with each other in a self-complementary manner and hetero complementary motifs that require different complementary motifs to be associated in a hetero complementary manner [89,103], as shown in Figure 4A-i,A-ii. At present, the latter method is widely used in designing more complex polymer networks because its cross-linking strength can be achieved by adjusting various complementary motifs. We summarized different physical cross-linking design schemes for constructing such supramolecular polymer networks, as shown in Figure 4.

In scheme B-i, since the supramolecular complementary motif can only self-assemble in a bivalent manner, at least three components are required to construct a cross-linked polymer. In scheme B-ii, if the supramolecular complementary motifs at the ends of multiple linear chains can form more than two association nodes, these motifs are cross-linked. In the B-iii scheme, the supramolecular linear chain can form cross-linking through the lateral stacking and aggregation of dynamic cross-linking points, and it can form a polymer network under the entanglement and entanglement between linear chains. In scheme C, the supramolecular motif is connected to the polymer backbone as a side chain, and even if the supramolecular motif is assembled in a bivalent manner, it will lead to cross-linking of the polymer chain. If a hetero complementary motif is used, cross-linking can be achieved by adding a small molecule cross-linking agent to the supramolecular cross-linked polymer (C-i) or by using a second polymer (C-ii) functionalized with a complementary supramolecular motif.

In certain conditions, the formation and collapse of the main chain type supramolecular network can lead to a relatively complete change of the polymeric materials, thus producing degradable or self-healing materials [104]. In the latter construction strategy, the binding effect between polymer chains can be flexibly adjusted to control the rapid response of materials in structure and performance. When the polymer backbone is composed of reversible non-covalent modules, the resulting supramolecular polymer is linear or cross-linked. Generally speaking, linear supramolecules will adopt homocomplementary or heterocomplementary motifs, while the construction of cross-linked polymer networks requires three—or even four view components [70,105]. For example, researchers often use multimeric cyclodextrins as physical cross-linking points to form more complex three-dimensional dynamic network structures [106]. On the one hand, the polymerization degree and molecular chain length of the main chain type supramolecular polymer can be artificially controlled by the process of induced polymerization and depolymerization. On the other hand, when the functional side chain is grafted to the polymer backbone through non-covalent bonds, the side chain can attach different types of functional groups to improve the rigidity of the polymer chain and the resistance to metal cations. Therefore, one advantage of side-chain supramolecular polymers is that functional monomers can be flexibly switched without damaging the static polymer backbone linked by covalent bonds.

In addition, if different non-covalent bonds are inserted in the main chain and the side chain, a highly functional dynamic polymer structure can be realized so that both the main chain and the side chain have supramolecular properties [107,108]. This multi-step preparation of supramolecular polymers driven by multi-non-covalent interactions helps to realize the multiple responsiveness of intelligent oilfield chemical materials to various stimuli or artificially controlled external factors in the reservoir. For example, Jiang Guancheng [109] constructed molecular aggregates with specific structures (XG-β-CD-S) through hydrophobic interaction, hydrogen bonding and host–guest interaction. The amphiphilic structure of the surfactant plays a good role in linking the hydrophilic group and the hydrophobic group. The hydrophilic head group and the strong polar acetyl group and pyruvic acid group on the xanthan gum branch chain are linked by hydrogen bonds. The hydrophobic part of cyclodextrin and surfactant is bound by the host–guest interaction. Therefore, driven by a variety of non-covalent interactions, the surfactant and β-cyclodextrin produced directional links between the main chains of xanthan gum, forming a relatively regular spatial network structure. In subsequent experiments, XG-β-CD-s shows shear thixotropy. Since different bonds respond to different environmental stimuli, the self-assembled system realizes both “salt response” and “thermal response”, forming a functional adjustable system that can be precisely controlled.

### 3.2. Polymerization Techniques in Oil Field Chemistry

Due to the limitations of economic cost and experimental conditions, supramolecular polymers as oilfield chemical materials cannot have complex and precise synthesis steps as biomedical materials. Therefore, we believe that the ideal supramolecular materials should have simple synthesis methods and can be customized for different construction sites. In the past decade, various supramolecular systems have been synthesized by different polymerization technologies. These technologies are reviewed in detail in this section.

#### 3.2.1. Free Radical Polymerization (FRP)

Free radical polymerization (FRP) is usually a polymerization technology that induces an initiator to initiate monomer activation to form monomer free radicals under the action of heat, light and radiation, and then, it forms long-chain free radicals through the chain polymerization mechanism to obtain polymers [110,111]. FRP monomers come from a wide range of sources, are easy to operate, have various synthesis processes, and have low industrial costs. Under the condition of removing oxygen and polymerization inhibitors, FRP can be carried out smoothly even in aqueous solution. Therefore, FRP, as one of the most important preparation methods in the production of oil field polymer, is widely used in various polymerization processes, including solution polymerization, suspension polymerization, lotion polymerization, etc. In the field of supramolecular polymers, most of them use FRP to prepare random coiled polymers.

Lingyu et al. [112] reported on the preparation of a supramolecular tackifier through FRP using an anionic monomer (2-acrylamide-2-methylpropanesulfonic acid) and cationic monomer (Dimethyldiallyl Ammonium Chloride), which significantly improved the dynamic plastic ratio and cutting capacity of drilling fluid. The results show that the structure of the supramolecular network depends on the composition of the initiator, the molar ratio of monomers and the concentration of each reaction component in the system. However, when the synthesized polymer chain contains hydrophobic groups, the mixing between hydrophilic monomers and hydrophobic monomers must be solved. Micelle free radical copolymerization is the most commonly used method to prepare polymer containing hydrophobic monomers. In order to introduce a large number of ethoxyl units and long-chain alkane hydrophobic units into the polymer backbone, Jiang Feng [113] used acrylamide-based fatty alcohol polyoxyethylene ether (hydrophilic monomer), petam (hydrophobic monomer) and r-gen259 (photoinitiator) to prepare salt-resistant supramolecular polymers in N_2_ atmosphere, and they added surfactants (SDS), using the so-called “surfactant micelle polymerization”. Hydrophobic monomers were solubilized in micelles formed by sodium dodecyl sulfate. With the decomposition of a photoinitiator, water-soluble free radicals were generated in the aqueous phase to initiate the polymerization of hydrophilic monomers. When the head of the chain growing molecular free radicals and the SDS micelle expand, the hydrophobic monomer will enter the micelle and introduce the short hydrophobic chain to form the hydrophobic micro region of the micro block structure. Then, the free radical heads leave the micelles, and such steps are repeated several times until the macromolecular free radicals stop. Due to the randomness of this process, hydrophobic micro blocks are randomly distributed in the polymer chain. Considering the influence of hydrophobic monomers on polymer solubility and the polymerization inhibition of surfactants, the proportion of each component should be strictly controlled to obtain the best supramolecular aggregates [113,114].

Up to now, the covalent polymerization of most supramolecular polymers in oilfield chemistry mainly involves FRP. However, since free radical polymerization is an uncontrollable polymerization process with slow initiation, rapid growth and rapid termination, the molecular weight of the polymerization product is uncontrollable, and the molecular weight distribution is wide. FRP cannot synthesize polymer networks with precise structures, such as block type supramolecular polymers and supramolecular star polymers. Of course, FRP is still a suitable choice for preparing supramolecular random copolymers without precise control of the molecular weight distribution and complex molecular structure.

#### 3.2.2. Reversible Deactivation Radical Polymerization (RDRP)

In FRP, the life of the growth chain is less than 1 s. At the end of the reaction, all polymer chains are “dead”. The polymerization reaction is not active/controllable, and the performance of the polymer cannot be accurately controlled. In 1956, Scwarc [115] first mentioned the concept of active polymerization in anionic polymerization. The so-called active polymerization refers to those polymerization reactions that do not have side reactions such as irreversible chain transfer or growth chain termination reaction. So far, many active polymerization systems have been developed to precisely control the segment distribution, end group control and molecular weight distribution of polymers, including atom transfer radical polymerization (ATRP), nitrogen oxygen regulated polymerization (NMP) and reversible addition fracture chain transfer polymerization (RAFT) [116,117]. The basic idea of reversible/inactivated radical polymerization (RDRP) is to realize the rapid switching of free radicals between dormant chains/active chains through the construction of highly active free radical reversible transfer or reversible termination inactivation reactions, so that the equal probability growth of polymer chains is dominant. Through RDRP, different monomers can be polymerized under mild conditions, and the polymer structure can be controlled at a high level. In addition, RDRP is not sensitive to water environments, which is a key advantage in oilfield industrial polymerization. Because the RDRP chain has a longer life, the second or even the third functional block can be added, and the end group with a complex structure can be grafted to prepare supramolecular polymers meeting specific functional requirements.

In the synthesis of supramolecular polymers, some supramolecular structures are complex (hydrogen bond ordering), the end group function is clear, the molecular weight distribution is narrow, and the synthesis method with strong design ability is required. Raft is more suitable for a variety of monomers than NMR or ATRP, especially olefin monomers with special structures, which can effectively control the molecular weight distribution of polymers [118,119].

As for synthesizing supramolecular polymers with complex topological structures, a typical example is that by using RAFT polymerization, we can well protect functional monomers or functional groups on raft reagents (such as –OH, –SO_3_). We can prepare terminally functionalized supramolecular polymers by first connecting the required supramolecular monomers to raft reagents and then introducing the functional groups of raft reagents into polymers. Muwang et al. [120] successfully synthesized base terminal functionalized hydrogen bonded block copolymers by using raft, as shown in Figure 1. In order to prepare the first block RAFT polymerization of polymer, hydroxyethyl adenine (A) and thymine (T) were first synthesized. 4-Cyano-4-(dodecylsulfonamidothiocarbonyl) sulfonamido valeric acid (dttcp) was used to react with modified nucleobases to obtain raft reagent dttcp-A/T, and then, the raft reagent was grafted onto oligomeric (ethylene glycol) methyl ether methacrylate (oegma). The obtained first block a-poegma continued to be polymerized with n-butyl methacrylate as a raft reagent, and finally, an amphiphilic block copolymer containing adenine and thymine at the end was synthesized. The self-assembly behavior of this short-chain block copolymer in aqueous solution was observed despite the highly competitive aqueous environment.

The RAFT polymerization method is also often used to design and synthesize star polymers. In 2010, Lortie and Bernard [121] designed a chain transfer agent with thymine and diaminopyridine; they synthesized linear polymers with thymine at the end by the polymerization of vinyl acetate units initiated by RAFT polymerization, and then, they synthesized supramolecular star polymers through mutual recognition between multiple hydrogen bonds. Subsequently, the group continued to use this strategy, first synthesizing different types of raft chain transfer agents containing thymine units, then initiating RAFT polymerization of different monomers, and finally assembling two different types of polymers together through molecular recognition to form AB_2_-type heteroarm supramolecular star polymers [122].

At the same time, as a multi-functional process for the synthesis of polymers with a multi-topological structure, raft can not only control the molecular weight of linear polymers but also allow us to adjust the distance between cross-linking points in cross-linked polymer networks to obtain more regular polymer networks and induce the dynamic networks to self-assemble [123], which is exactly what is needed for the construction of ordered supramolecular networks.

#### 3.2.3. Post-Polymerization Modification

Post-polymerization modification is an effective way to replace small molecule monomers to prepare functional polymers by further chemical modification of the prepared polymers. Many functional polymers have been prepared by different polymerization methods and post-polymerization modification. However, there are few reports on the post-polymerization modification of synthetic oil displacement polymers. We believe that with the increasing demands on the versatility of modified polymers in the oil field construction site, the polymerization modification will receive extensive attention in the next few years. At present, the application of post-polymerization modification mainly focuses on natural oil displacement polymer. Yingruibai et al. [124] grafted hydrophobic long-chain alkane (1-bromosunane) onto the hydroxyethyl side chain through post-polymerization modification to obtain hydrophobically associated hydroxyethyl cellulose (hahec). When the critical association concentration was reached, the formation of supramolecular aggregates driven by hydrophobic interactions was observed by atomic force microscopy. Similar to Jiang’s idea, Bingwei et al. [108] chemically modified xanthan gum through non-covalent bonds such as hydrogen bonds and host–guest interaction to build a new self-assembled biopolymer network suitable for harsh reservoir conditions. After the alkaline hydrolysis of xanthan gum in brine, under the action of hydrogen bonds and van der Waals force, the branch chain is connected with the hydrophilic head of the surfactant, and the hydrophobic double tail is encapsulated in the lumen of β-cyclodextrin. It has been shown that such SAP dynamic networks have good temperature and salt resistance and exhibit thixotropy under high-speed shear.

Zou et al. [125] used N-isopropylacrylamide and vinyl ferrocene (Fc) to copolymerize to obtain ferrocene functionalized PAM, but its molecular weight is only about 2000, and the grafting rate and yield are low. As a result, the ideal rheological properties cannot be obtained. Jiang Zhiping [126] directly alkylated HPAM in dimethyl sulfoxide to graft ferrocene functional groups. The results of NMR calculation showed that the grafting rate of the FC group (1.3 mol%) was significantly improved.

## 4. Characterization Methods of Supramolecular Polymers

A series of new supramolecular oil displacement materials have been developed. However, traditional polymer characterization techniques are still used to study the structure of supramolecular polymers in most reports, and the study of the supramolecular self-assembly mechanism is rarely involved. On the one hand, improving oil recovery is the main purpose of using supramolecular polymers. After the formation of a supramolecular three-dimensional network is observed by using conventional characterization technology, it is sufficient to explain the contribution of non-covalent physical cross-linking to the viscoelasticity of polymer solution. On the other hand, the dynamic characteristics of non-covalent bonds make it difficult to characterize them. Changes in pH, temperature or environmental stimuli brought by characterization equipment usually affect the original molecular structure of self-assembled modules [127,128]. In this section, we summarized various technologies for characterizing supramolecular polymers, including characterizing the supramolecular polymer chain itself and its self-assembly state, and we discussed the advantages and disadvantages of these technologies. Although some technologies are used in biomedical and material engineering fields, we believe that if they are extended to the field of oilfield chemistry, it will open up a new research direction for EOR research.

### 4.1. Rheological Characterization

Similar to conventional polymers, the comprehensive and systematic rheological characterization of supramolecular polymers is required in EOR research, which includes the following. (1) First, there is the determination of apparent viscosity. Viscosity is an important parameter to characterize the resistance of the fluid against flow. It should be noted that the viscosity of the polymer changes with the shear rate or the stretching rate, and it generally belongs to pseudoplastic fluid. The viscosity measured by the rheometer is the apparent viscosity under the specific strain rate. (2) Second, there is a shear thinning behavior test. The water-soluble polymer solution deforms under the shear force. With the increase in shear rate, the viscosity of the polymer solution decreases gradually. (3) Third, there is the viscoelasticity test. Viscoelasticity makes a great contribution to the macro/micro oil displacement efficiency of polymers. It is usually necessary to scan the dynamic strain of polymer solution to test the change curve of elastic modulus (G′) and viscosity modulus (G″) with strain and viscosity loss. (4) Fourth, there is the thixotropy test. With the increase in shear rate, the network structure of polymer was destroyed. Subsequently, the shear rate was reduced without interference, and the polymer solution structure gradually recovered. Due to the reversibility of non-covalent bonds, the shear thixotropy of dynamic networks of supramolecular polymers is obviously different from that of static covalent networks, as shown in Figure 5A,B.

Another significant difference in rheological characterization between supramolecular polymers and conventional covalent polymers is that transient networks usually have complex dynamics, which may lead to wrong theories and explanations based on limited rheological test results. At this time, it is necessary to combine a variety of viscoelastic techniques, spectra and other experimental evaluations. A notable example is the polymer with adamantane/cyclodextrin as the supramolecular unit. In Takuya’s research [25], the viscoelasticity of this supramolecular polymer with a clear structure comes from the contributions of various cross-links, and it is not limited to the host–guest interaction between cyclodextrin and adamantane. As shown in Figure 4C–F, the analysis of dynamic viscoelastic results (Figure 5C) shows that the flat modulus at the low-frequency limit was equal to the rubbery flat modulus of the polyacrylamide solution without a cross-linking structure, indicating that the viscoelasticity in this range is caused not by non-covalent cross-linking but the entanglement of the main chain. Then, in HPAM containing only β-CD (Figure 5E), with the increase in β-CD content, the terminal relaxation time is prolonged, and the plateau modulus remains unchanged at high frequency. Similar to the cross-linked polymer, the β-CD molecule becomes the cross-linking point of HPAM without a guest. The 2D NMR results (red squares in Figure 5D) show that the β-CD molecule is related to the protons in HPAM backbone, which indicates that the backbone penetrates β-CD to form a rotaxane structure. The above structure proves that there are three kinds of cross-linking in the host–guest system: static entanglement of HPAM backbone, static cross-linking of rotaxane structure and dynamic cross-linking of host–guest complex.

In addition, combining with rheological data, a physical simulation experiment, and a screen coefficient test, the matching relationship between polymer hydrodynamic size and reservoir permeability and pore throat can be determined. This has important reference significance for the percolation characteristics of supramolecular polymers in porous media and the self-assembly process of supramolecular networks in the pore throat.

### 4.2. Structural Characterization of Supramolecular Polymers

For scientists in the oil field, the common means to study the molecular structure and three-dimensional network shape of any polymer is to combine microscope technology and spectral analysis technology, such as transmission/scanning electron microscope (TEM/SEM), infrared spectrum analysis (IR), nuclear magnetic resonance spectrum (NMR), etc. We summarize the different technologies used to study the polymer for oil displacement in recent years. Using this information, we can design a characterization system suitable for different supramolecular polymer structures.

Atomic force microscope (AFM) is another widely used high-resolution scanning probe microscope, which is inexpensive and can directly observe the morphology of supramolecular polymers. Compared with TEM/SEM, the biggest advantage of AFM is that it can easily obtain the three-dimensional sample profile and surface phase distribution information deposited on the surface, and it can avoid the problems of insufficient contrast of supramolecular system and sample drying treatment [130]. Zhong et al. [131,132] synthesized salt-resistant supramolecular polymer (PPSA) through the copolymerization of acrylamide, alkyl-fused aromatic unsaturated monomer and amps. As shown in Figure 6, in order to study the effect of supramolecular aggregates on the three-dimensional morphology of polymers, the atomic force morphologies of polymer solutions with different concentrations were obtained. It is obvious that the diameter of the molecular chain bundle increases with the increase in the concentration. When the polymer concentration is 1.5 g/L, due to the electrostatic repulsion and hydrophilicity of the polymer chain, the molecular chain remains extended and the molecular arrangement is more regular. At the same time, the increase in hydrophobic interaction is conducive to the formation of a continuous and structured unique supramolecular structure.

Different from AFM, fluorescence microscopy as another microscopy method involves the characterization of supramolecular systems using the interaction of fluorescent probes with polymers. Ans is a well-known fluorescent probe, which is usually used to detect the presence of hydrophobic micro regions in intramolecular and intermolecular [4]. Roy et al. [133] studied the association behavior of hydrophobic-modified xanthan gum (X_29_C_8_) through an ANS fluorescence microscope. It was observed that the fluorescence intensity of X_29_C_8_ increased more than that of the xanthan gum precursor at the same concentration, which confirmed the existence of hydrophobic microdomains generated by supramolecular interactions between xanthan gum. On the basis of fluorescence microscopy, a laser-scanning confocal fluorescence microscope (lscfm) can rapidly scan and image the samples containing fluorescent substances point by point, line by line and surface by surface [134]. In polymer materials, lscfm can be used to characterize polymer morphology, surface and interface structure.

Scattering technology is another reliable tool for characterizing supramolecular polymer systems. Light scattering is a mature method to determine the molecular size and morphology of samples. The classical static light scattering is used to obtain the weight average molecular weight Mw, radius of gyration RG, and the second virial coefficient A_2_ of the polymer through the relationship between the time average scattering light intensity and the angle and concentration. It should be noted that the curve of the covalent polymer is linear, and the dynamic characteristics of the supramolecular polymer lead to the change of A_2_ [135] with the molar mass concentration. In order to avoid this problem, an effective strategy is to use a chain stopper to temporarily control the degree of supramolecular polymerization [136,137]. Dynamic light scattering is used to study the size and distribution of aggregates formed in supramolecular polymer solution by measuring the change of light intensity with time. For example, Hao and his colleagues [138] constructed a Y-type supramolecular polymer by the inclusion interaction between β-cyclodextrin and azobenzene. The self-assembly process was monitored by DLS, providing auxiliary evidence for TEM, fluorescence spectroscopy and other testing methods. Figure 7 shows TEM and DLS results of PNIPAM-(2CD-2MPEG) solution at different temperature points. From the TEM results (Figure 7A–E), with the increase in temperature, the self-assembled morphology gradually changed from dot-like micelles (20 nm) to core-corona-structured micelles (175 nm). When the solution temperature decreased to 20 °C, the self-assembled morphology gradually changed to point micelles again. Accordingly, the DLS results show that there is a similar reversible self-assembly process (Figure 7 F–G). Light scattering has become an important means to study the microstructure of supramolecular polymers due to its advantages of small sample interference, simple operation and fast analysis speed.

### 4.3. Study on the Mechanism of Supramolecular Self-Assembly

The dynamic network of supramolecular polymers based on non-covalent cross-linking not only endows them with unique properties but also makes it very difficult to study the dynamic mechanical properties and self-assembly mechanism. In order to better understand and deepen the understanding of supramolecular systems, the study of supramolecular interactions is particularly important. The supramolecular assembly characterization techniques in some frontier fields are discussed, and some self-assembly studies in oilfield chemistry are illustrated.

**Commonly used spectroscopy techniques.** In most reports on self-assembled supramolecules, UV/visible spectra, fluorescence or circular dichroism spectra, and nuclear magnetic resonance spectroscopy are often used. If the supramolecular system contains spectral active substances, we can observe the aggregation state of supramolecules, calculate the binding constant and analyze the complex conformation of supramolecular aggregates by detecting the spectral changes (excitation/absorption) caused by chromophores in the system. Because of their relatively mature technology, all these characterization tools are used by most laboratories to quantitatively study the supramolecular effects in solution. Using these techniques, spectra were recorded from solutions containing monomers/polymers at known concentrations in the desired solvent. By taking the additional spectra as a function of concentration or temperature, the transition from the polymer species to its monomer state can be easily observed [127]. Therefore, they are very useful in characterizing supramolecular systems.

**Single molecule force spectrum based on atomic force microscope.** Single molecule force spectroscopy (SMFS) based on atomic force microscopy (AFM) is a new and effective method for studying intramolecular or intermolecular interactions in macromolecules, which can realize the manipulation of single molecules and characterize the mechanical behavior of single molecular chains [127,139,140]. So far, AFM single molecule force spectroscopy has been successfully used to characterize long-chain supramolecular polymers with certain stability. For example, Zou et al. [141] modified the AFM probe and substrate with the 2-urea-4-pyrimidine ketone group (UPy), forming a bridge structure between the probe and substrate through the hydrogen bond between the UPy dimer. Then, the polymerization degree information of the supramolecular polymer was calculated according to the fracture length and the UPy unit length on the tensile curve. Based on this research, Jingsi Chen et al. [79] tethered different alkylene long chains to UPy dimers and characterized the mechanical binding strength and kinetic parameters of hydrogen bonds in water by combining SMFS and molecular dynamics (MD), as shown in Figure 8A. The SMFS experiment revealed that the hydrophobic effect of alkylene groups enhanced the binding strength of upy hydrogen bonds at the single molecule level (Figure 8B,C). At the same time, the MD simulation results showed that with the increase in the number of carbon atoms in the hydrophobic long chain, the binding free energy increased by 8.7 kJ/mol, which further proved the effectiveness of enhancing hydrogen bonds through a hydrophobic interaction in the water environment.

**Low-temperature transmission electron microscope.** In many studies, the samples characterized by TEM and SEM usually need to be dried. It is well known that sample dehydration may change the material structure of soft materials, which is dangerous to study the overall supramolecular structure and local specific details of aggregates [143,144]. In recent years, with the improvement of supported membrane technology and the development of rapid freezing technology (vitrojet technology, spotton Technology), direct imaging low-temperature transmission electron microscopy (cryo TEM) has become a powerful characterization tool for obtaining direct information at the supramolecular level. The original structure information and molecular conformation of the samples in the natural aggregation state can be maintained by low-temperature freezing treatment. Capturing metastable and short-lived intermediates is often the key to understand and prove the supramolecular self-assembly mechanism and complex structure [145,146]. Through the recent work of our team [142] in revealing the micro self-assembly mechanism of surfactants, the importance of studying the self-assembly of soft materials by using freezing transmission electron microscopy is emphasized. In the polar or nonpolar mixtures of sodium dodecylbenzene sulfonate (SDBS)/industrial betaine with anisole, white oil, etc., the self-assembly morphology transition and molecular solubilization sites of micelles in the mixed micelles were confirmed by freezing electron microscopy. Specifically, as shown in Figure 5D,E, the addition of anisole to SDBS rod-shaped micelles (diameter = 23 nm, length = 100 nm) with lower concentration (0.05 wt %) will induce rod worm-like transition. Driven by electrostatic interaction and hydrophobic effect, the micelles grow into linear worm-like micelles (diameter = 80 nm, length = 350 nm), with a volume expansion of 40 times, while the mixed micelles of kerosene and sodium dodecylbenzene sulfonate are in the transition state between rod worm-like (Figure 5F,G), which is consistent with the DLS results. At the same time, through the Cryo TEM image, we can accurately determine the sites of anisole and kerosene in the aggregates through the distribution of black spots representing solubilized molecules. In conclusion, the low-temperature temperature is the key evidence that the solubilization ability of micelles strongly depends on the molecular solubilization sites and the morphology of swollen micelles. However, among the 242 self-assembly research reports related to enhanced oil recovery, only 12% of the research literature used low-temperature transmission electron microscopy. We strongly recommend that this technology become a standard technology for chemical supramolecular characterization in oil fields.

### 4.4. Summary

Revealing the nature and dynamic process of the interaction in supramolecular bodies and the influence of external environmental conditions on the stability of a single action site at the molecular level is conducive to enhancing people’s understanding of the essence of supramolecular assembly behavior and the relationship between the functional properties of supramolecular and the building elements. In order to fully study the self-assembly mechanism, we need to combine the correct technology applicable to the supramolecular system, flexibly use various characterization methods, and consider the binding constant of supramolecular interaction, instrument sensitivity, molecular physicochemical properties and other factors. This complex and tedious characterization work is necessary for the challenges brought by the new self-assembled supramolecular system designed in EOR research.

## 5. Application Progress of Supramolecular Polymers in EOR

### 5.1. Enhance Oil Displacement Efficiency

EOR technology is mainly used to obtain more oil and gas resources by expanding the sweep coefficient and improving the microscopic oil displacement efficiency [2,3]. The supramolecular polymer based on the theory of “supramolecular chemistry” and “adaptive chemistry” has received the attention of oilfield workers. Introducing non-covalent effects such as electrostatic interaction, host–guest interaction and hydrogen bonds into the oil displacement polymer is expected to improve the rheological performance of the oil displacement agent and greatly improve the oil displacement efficiency under the increasingly harsh reservoir environment.

Electrostatic interaction has long been used to enhance the temperature and salt resistance of polymers. Researchers usually copolymerize acrylamide with anionic monomers and cationic monomers to obtain ionic polymers with different structures so as to reduce the irreversible degradation of polymer chains caused by high temperature and salinity. Through experiments, the influence of the introduction of electrostatic interaction on polymer-enhanced oil recovery was explored. In recent years, the research on electrostatic interaction is not limited to simply improving the rheological properties of polymers. Scientists have found that the charge properties of polymer chains or small molecules can be indirectly controlled by controlling the pH or CO_2_ concentration of aqueous solution so as to accurately control the self-assembly process of supramolecular systems. For example, Yegin et al. [147] used electrostatic interaction as a driving force to form worm-like micelles through the self-assembly of small molecules between amino amide and maleic acid, and they realized high-viscosity fluid with pH cyclic response. They proposed to solve the problem of injectability by adjusting pH to control viscosity. Xiong et al. [148] reported a CO_2_-responsive smart fluid based on a supramolecular assembly structure from vesicles to worm-like micelles, which has shear thinning and self-healing characteristics, providing a new strategy for the design of CO_2_-responsive smart fluids and promoting their application in EOR. This intelligent response fluid forms vesicles under the driving of hydrophobic effect. In the presence of CO_2_, the single-chain weak cationic surfactant changes to the Gemini cationic surfactant, and its self-assembled structure changes from vesicles to worm-like micelles. The micelles are entangled to form a three-dimensional network structure that is “polymer like”. Therefore, the zero shear viscosity of the solution is increased by more than four orders of magnitude, as shown in Figure 9A.

Host–guest supramolecular oil displacement systems have been widely used in the field of enhanced oil recovery in recent years, significantly improving the temperature and salt resistance of polymers. As early as 2012, Zou et al. [35] developed a new polymer (AM-co-A-β-cd-co-AE) by copolymerizing acrylamide, modified cyclodextrin and hydrophobic monomers The non-covalent “interlocking effect” between β-Cd and the hydrophobic monomer endows the polymer with a solid and compact three-dimensional network, which has potential application prospects in high-temperature and high-mineralized oil fields. In many cases, in the study of enhanced oil recovery by β-CD and host–guest effect, the temperature and salt resistance of the polymer have been significantly improved. Interestingly, some cyclodextrin groups even exhibit the unique behavior of “salt thickening” [35,37,95]. As shown in Figure 9B, on the one hand, the introduction of macromolecular rigid monomers such as cyclodextrin brings about large steric hindrance, which can improve the hydrodynamic volume of polymer and avoid the molecular chain curl caused by high temperature and high salinity to a certain extent. On the other hand, when the temperature rises, the inclusion of the hydrophobic cavity of cyclodextrin with guest molecules is promoted, and the non-covalent cross-linking between polymer chains is enhanced. In addition, the increase in metal cation concentration in the aqueous solution will increase the polarity of the solution, and the host–guest structure is enhanced under the driving of hydrophobic effect. Therefore, it can be found that the viscosity of the polymer increases to the maximum value with the salinity and then decreases.

In addition, the supramolecular system based on hydrogen bonding has also been gradually applied to tertiary oil recovery. Zhao et al. [149] synthesized a polymerizable monomer GF (N-dimethylaminoethyl glucoside) with glucose and then polymerized with acrylamide to obtain an environmentally friendly oil displacement polymer (Figure 9C). Under the harsh environment of 200 g/L mineralization, the system can still maintain high viscosity. These excellent properties are attributed to the formation of intermolecular hydrogen bonds and the complexation of metal cations with glucoside units. Our team refers to the DNA structure and biological supramolecular gel network [150,151], and it introduces adenine and thymine into the polymer chain. As shown in Figure 9D, it hopes to build a supramolecular dynamic network through multiple hydrogen bond arrays between bases. Although the polymer does not show the expected responsive dynamic behavior, in the core injection evaluation, due to the existence of reversible hydrogen bonds in the dynamic network, the polymer has lower injection pressure in low-permeability core. Of course, supramolecular materials with hydrogen bonds as the main driving force still have a long history of development to be truly applied in the field of enhanced oil recovery. Since the strength of hydrogen bonds is greatly weakened in polar water environments, other cooperative hydrogen bonds including hydrophobic force and complexation are required to realize the reliable supramolecular structure of materials [152], which increases the difficulty in molecular design, monomer synthesis and polymerization.

### 5.2. Auxiliary Gas Drive

Low-permeability reservoirs are faced with such problems as difficult water injection, high injection production ratio, slow pressure transmission and insufficient formation energy. Gas injection development can effectively solve the problem of injection. However, due to the development of natural/artificial fractures, the macro and micro heterogeneity of the reservoir is strong, resulting in serious gas channeling in the CO_2_ oil displacement process of low permeability reservoirs, and the sweep efficiency is reduced. In recent years, supramolecular polymers have been applied in controlling CO_2_ mobility, playing a good role in supplementing formation energy and promoting the development of gas injection profile control materials.

Supramolecular polymers have been widely studied in the field of water shutoff and profile control. Gao et al. [153] studied a kind of supramolecular aggregate formed by low molecular weight phenolic resin. This composite aggregate driven by hydrogen bonds can fully block the 1.2 µm core pore membrane, and the microstructure of the profile control system has dynamic changes in porous media, which can realize “blocking propagation re blocking”. Bai et al. [154] cross-linked a self-lubrication supramolecular network, polyacrylamide network and sodium polyacrylate network to obtain a high-strength gel with self-lubricating function and good deep profile control ability.

In gas injection profile control, researchers focus on the research of CO_2_-responsive polymer. CO_2_ self-responsive materials have high applicability in CO_2_ flooding and have good research prospects. While meeting the profile control capability, the CO_2_-responsive performance of the material can be achieved by introducing CO_2_-responsive functional groups (such as amidine, amine, guanidine and other basic groups).

Pu et al. [155] first synthesized polyacrylamide gel (PPSA) with methylene bisacrylamide as the cross-linking agent and then polymerized and cross-linked CO_2_-responsive monomer, acrylamide and the first network in situ to form an interpenetrating network polymer gel (IPN-PPSA), revealing its response mechanism (Figure 10A). In the indoor EOR experiment, the injection pressure of IPN-PPSA increased from 5.0 kPa before contacting CO_2_ to 342.0 kpa. The injected CO_2_ successfully infiltrated into the matrix to displace the crude oil, and the recovery rate in the dense core (0.85 md) increased by 8.2%. Luo et al. [156] added synthetic triblock tackifier polymer (f127-g-pdmaema) to water slug to control gas channeling during water–oil displacement. It is worth noting that the copolymer shows the synergistic thickening ability of heat/CO_2_ and forms a stable gel above the gel temperature. The experimental results show that the thermal thickening originates from the double LCST characteristics of the main chain and the side chain of the copolymer, and the CO_2_ responsiveness is attributed to the CO_2_-responsive branch chain (pdmaema) introduced in the side chain. When CO_2_ is dissolved in the aqueous solution, on the one hand, a large number of tertiary amine groups are protonated in the acidic aqueous environment, and the molecular chain of pdma-ema is extended due to the electrostatic repulsion force, resulting in expanded micelle aggregates, and the viscosity of the system is greatly increased; On the other hand, the branched chain becomes highly hydrophilic due to protonation, which changes the solubility of the LCST group and enhances the thermal thickening effect of the polymer. In the core displacement experiment, the copolymer aqueous solution and CO_2_ were alternately injected into the high-permeability (610 md) and low-permeability (76 md) cores, as shown in Figure 10B. With the increase in injection volume, the injection pressure difference increased continuously, indicating that CO_2_ and high temperature increased the viscosity of polymer slug. The graft copolymer blocks the high permeability layer, improves the core heterogeneity and alleviates the gas channeling of CO_2_.

By virtue of self-assembly and self-responsiveness, supramolecular polymers are expected to adapt to different permeability in heterogeneous reservoirs, achieve step-by-step profile control, and make up for the shortcomings of existing materials. However, field application has not been reported. In addition, the self-response performance is regulated by the self-assembly of molecules by the protonation of basic groups. This kind of CO_2_ response system is essentially pH response. The CO_2_ response mechanism based on BRønsted acid–base theory has some shortcomings. For example, the weak acidity of CO_2_ leads to chemical inertia and a limited reaction environment. In recent studies, some people introduced frustrated Lewis pairs (FLPs) into polymers [157,158,159], so that CO_2_ molecules can act as “physical cross-linkers” to bridge polymers to achieve CO_2_ solution gel response, as shown in Figure 10C. The influence of weak acidity and insufficient protonation of CO_2_ is reasonably eliminated.

## 6. Challenges and Future Directions

### 6.1. Comprehensive Comparison between Conventional Polymer Materials and Supramolecular Polymers

In order to show the differences between conventional polymer materials and supramolecular polymers as comprehensively as possible, Table 1 lists the visual comparison between them item by item, showing the advantages of supramolecular polymers in EOR research and the limitations of supramolecular polymers over conventional polymer materials.

In terms of material source and cost, supramolecular polymer materials used to construct excellent oil displacement performance have a wide range of sources. Many functional monomers are from natural raw materials, but some monomers need to be chemically modified, and the cost needs to be reduced. However, conventional oil displacement polymers based on limited materials (petroleum based monomers) have low cost. From the perspective of synthesis process, there are various synthesis methods involving supramolecular polymers, including FRP, raft, NMR, supramolecular bonding polymerization, coordination/synergistic polymerization, etc. Sometimes, it is necessary to carry out complex chemical modification on monomers and polymer precursors to meet different supramolecular properties. Polymer synthesis methods and strategies are relatively simple, and most polymer structures can be realized by FRP. From the perspective of characterization, in addition to the necessary structural characterization and rheological tests, we need to study the self-assembly behavior of supramolecular polymers, observe the local details of supramolecular aggregates, and determine the binding constants to obtain kinetic data. Most of the reports on conventional polymers can obtain convincing evidence on polymer structures through FT-IR and 1HNMR. From the perspective of a polymer network structure, supramolecular polymers are mainly reversible dynamic networks based on physical non-covalent cross-linking. On the contrary, conventional polymers are permanent and static networks based on chemical covalent cross-linking. From the perspective of environmental compatibility, supramolecular polymer raw materials have a wide range of sources, natural polymer precursors and bio-based monomers can be used, and non-toxic physical cross-linking agents can be used. Therefore, it shows good biocompatibility and biodegradability [169]. The raw materials of conventional polymers (acrylamide, chemical cross-linking agent, etc.) may pollute the reservoir environment. From the perspective of stimulus response, due to the dynamic characteristics of non-covalent interactions, supramolecular polymers can respond to various physical, chemical and biological factors, such as light, pH, temperature and ion concentration. However, the conventional polymer network cannot recover after the molecular structure is forced to change due to environmental factors. From the perspective of viscosity-increasing ability, the viscosity of conventional polymer solution is related to the molecular weight [170]. High molecular weight polymers usually have good viscosity-increasing ability, but this will bring about injectability problems. The cross-linking of a supramolecular polymer network is due to the interaction of physical cross-linking and polymer chain entanglement, which can exhibit high viscoelasticity at relatively low molecular weight. From the perspective of resistance to the harsh environment of the reservoir, many non-covalent interactions are insensitive to temperature and metal cations, and supramolecular polymers have good temperature and salt resistance. At the same time, the three-dimensional network of the supramolecular polymer is destroyed after high-speed shear, and the supramolecular units re-assemble when flowing through the space with a low shear rate to reconstruct the three-dimensional network of the supramolecular polymer [165,171]. This recombination phenomenon gives the supramolecular polymer excellent “shear recovery” characteristics. However, conventional polymers have always been faced with the problem of polymer curling and precipitation under high temperature and high salinity. In addition, conventional polymer chains are stretched in the underground, and the rheological properties after shear cannot meet the requirements of displacing crude oil. From the perspective of injectability, based on the dynamic network and stimulation response function of supramolecular polymers, the pumping pressure during injection can be reduced by controlling the viscosity. Compared with supramolecular polymers, we need to fully consider the injectability and molecular weight of conventional polymers, because polymer molecules are directly related to displacement viscosity. From the perspective of reservoir applicability, the existing polymers are generally used in medium and high-permeability reservoirs (50–10,000 md), t < 85 °C, K < 60,000 mg × L^−1^ [41,172,173], while supramolecular polymers have strong applicability and can cope with most harsh reservoir environments.

### 6.2. Challenges Caused by Supramolecular Polymers in EOR

Supramolecular polymers are different from traditional polymer materials in many aspects. The replacement of oil displacement materials may bring a series of challenges, including synthesis, characterization, field application and construction cost. These new problems limit the development of supramolecular polymers in EOR and other oil field chemical engineering fields. Our main concerns about supramolecular polymers are as follows:

#### 6.2.1. The Contradiction between Conventional Characterization Strategies and Supramolecular Polymers in Oilfield Chemistry

Supramolecular polymers are dynamic networks constructed by one or more non-covalent interactions, which often exhibit unexpected new properties, including shape memory, self-healing and special responses to environmental stimuli. In addition, there are various supramolecular characterization tools in the traditional application field, and each method has its advantages and disadvantages. Therefore, it is necessary to formulate appropriate characterization strategies according to the structure and unique performance of supramolecular polymers. We strongly recommend that we use two or more kinds of experimental technical means, adopt the characterization tools in the frontier field, and combine a variety of data from different angles to obtain convincing conclusions. For example, some researchers study the percolation law of supramolecular soft materials through microfluidic models [174,175]. Similarly, through the specific combination of fluorescent dyes and supramolecular units, we can simultaneously study the percolation law of supramolecular polymers in porous media and the self-assembly behavior of aggregates. Some characterization methods may no longer be applicable to dynamic polymers. Under the condition that the chemical structure of supramolecular polymers is not affected, physical methods can be considered to freeze the supramolecular polymers to characterize their static structures. With the rapid development of supramolecular chemistry in the field of enhanced oil recovery, we need to develop new experimental techniques for supramolecular oil field chemical materials.

#### 6.2.2. Precise Control of Structure and Performance

In all construction strategies, the density of chemical/physical cross-linking points has a great impact on the three-dimensional network and viscoelasticity of the system. For traditional macromolecular polymers, the cross-linking process is directly controllable, because the covalent macromolecular cross-linking network is related to the ratio of the introduced monomer and cross-linking agent or the degree of cross-linking reaction [176,177]. By controlling the covalent effect to cross-link between polymer chains, the molecular weight, polymer network and rheological properties can be precisely controlled, thus giving the polymer the expected oil displacement performance in EOR. It is difficult to predict and control the cross-linking density, self-assembly process and structure of supramolecular polymer systems cross-linked by non-covalent interactions, because these cross-linking points are dynamically reversible and usually in the balance of destruction and recombination. The cross-linking strength and network structure of non-covalent interaction may depend on environmental factors such as temperature, pH, ion concentration, or be affected by self-assembly process and network formation kinetics. Therefore, the self-assembly process and dynamic network that cannot be controlled accurately further limit the synthesis of supramolecular polymers with ideal structures. In order to solve this problem, we envisage using the post-polymerization modification strategy to synthesize supramolecular polymers with controllable molecular weight and functional group distribution to a certain extent.

#### 6.2.3. The Application of Supramolecular Polymer in Oil Field Is Limited

Because the versatility of supramolecular polymers often requires complex supramolecular structures, it is difficult to generate supramolecular polymer networks with ideal structures by simple synthesis methods. Only through the special chemical modification of functional monomers and polymer precursors can complex supramolecular networks be synthesized. Simple structural characterization and rheological testing of supramolecular polymers are far from enough. We need a variety of characterization methods to study the mechanism of supramolecular self-assembly and deepen our understanding of the mechanism of “supramolecular crude oil rock”. However, this is not compatible with the characterization strategy of traditional polymers. The complex synthesis process and a large number of characterization tests of supramolecular polymers usually lead to high development costs, which hinders the large-scale commercial production of supramolecular polymers. At the same time, it is difficult to achieve crude oil production at a low cost in the oil field.

The effective period of polymer flooding is generally more than half a year. The polymer is in the formation environment for a long time, so the polymer should have long-term stability in the formation. The transient three-dimensional network of supramolecular polymer is formed by dynamic cross-linking with limited life, and it is formed by self-assembly in a time scale of several minutes to dozens of days, showing significant viscoelasticity. In the formation, due to long-term high temperature, increased metal cation concentration, tensile shear and other factors, supramolecular polymers based on non-covalent interaction show different viscoelastic behavior from conventional polymers at different aging times. For example, in the amphiphilic supramolecular system reported by Zhang XF [90], the viscosity of the system increased from 118 to 1201 mPa·s in 30 days with the extension of aging time. Analysis shows that this is due to the enhanced electrostatic interaction between the main chains of HPAM. Hence, it is necessary for us to reveal the aging mechanism of different supramolecular networks at the molecular level to analyze the working time and durability of supramolecular polymers in the formation, but this increases the research difficulty and workload.

### 6.3. Future Directions

#### 6.3.1. Combination of Covalent and Supramolecular Polymers

The application of conventional covalent polymers in various reservoirs is limited, while supramolecular polymers show excellent oil displacement performance in EOR research. However, this does not mean that we can completely abandon the covalent polymer network. On the contrary, we can make use of the covalent polymer network to make up for the defects of the application of supramolecular systems in some reservoirs [178,179]. While non-covalent non-supramolecular polymers are endowed with dynamic characteristics, their polymer structures are also susceptible to changes due to environmental factors such as temperature and pH, so they are difficult to be applied to occasions requiring high mechanical properties and stability. For example, in the gel channeling sealing system, in the face of the high-temperature and high-pressure environment of the reservoir, it is required that the channeling sealing system has excellent mechanical properties to meet the gas-channeling plugging ability. In order to facilitate injection and prevent plugging of the formation, the dynamic characteristics of the polymer network are indispensable. The polymer research on covalent supramolecular coordination in this field has been mentioned in Section 4.3. Pu et al. [155] combined a CO_2_-responsive “IPN” supramolecular network with a high-strength PAM network (PAASP) to prepare new CO_2_-responsive preformed gel particles with an interpenetrating network. The protonation reaction is initiated after the IPN supramolecular network is contacted with CO_2_. The hydrophilicity change and electrostatic repulsion force make the polymer coil expand, and more water molecules enter the gel to increase the particle size, which makes the IPN-PAASP gel have CO_2_ response performance. At the same time, the PAASP covalent polymer network, as the protective scaffold of the IPN network, endows IPN-PAASP with excellent gel strength and makes it difficult for CO_2_ to break through the high permeability layer. The CO_2_ responsiveness enables IPN-PAASP gel to realize step-by-step profile control in reservoirs with different permeability through its own gel strength and particle size. In the review of Zhang and Stupp [105,179], the structure and performance advantages of covalent supramolecular polymers are discussed in more detail. We believe that this new type of polymer material is expected to overcome the inherent defects of a single polymer system and promote the vigorous development of polymer oil displacement materials.

#### 6.3.2. Development of Related Synthetic Processes and Characterization Techniques

In order to realize large-scale application, the ideal supramolecular polymer should have a simple synthesis method, easy molecular structure design, and be able to be widely used in oil fields. It is necessary to develop synthetic processes matching supramolecular polymers to obtain complex and accurate polymer structures with more commercial value. In order to study the oil displacement application potential of supramolecular polymers, it is necessary to establish a convenient and reliable structural characterization scheme. In addition, it is urgent to improve feasible self-assembly mechanism research strategies to further deepen the understanding of supramolecular oilfield chemical materials. Due to the urgent need of oil field construction, the cost of supramolecular polymers should be kept at a low level without damaging the reservoir and polluting the environment.

#### 6.3.3. Extending Complex Polymer Network

Supramolecular polymers with more complex topological structures can be designed and synthesized in the future. Although linear polymers play a dominant role in polymer flooding projects, simple linear structures should not be a constraint on the versatility of supramolecular polymers. With the development of synthesis technology and characterization technology, we can design topological structures such as multi-arm star polymers, hyperbranched polymers, tree networks and comb polymers. Replacing chemical cross-linking points in these complex polymer networks with non-covalent physical cross-links will certainly bring more attractive rheological properties and unexpected interesting dynamic properties.

Under different reservoir conditions, the performance of supramolecular polymers may depend on both static and dynamic characteristics, and the selection of monomer units and cross-linking agents also depends on polymer chain design, synthesis and non-covalent cross-linking points. At the same time, due to the different formation mechanisms of three-dimensional networks, the design ideas and synthesis methods of polymer and supramolecular self-assembly systems are very different, and it is impossible to predict the supramolecular polymer networks with complex structures based on the precursors and functional groups. Therefore, we need to fully consider the influence of supramolecular bonds and functional monomers on self-assembled networks. For a complex topological network structure, a chemical click function can be added after the network is formed for chemical post modification of the precursor network.

## 7. Conclusions

In this review, we discussed the reason and significance of studying supramolecular polymers in the field of enhanced oil recovery, summarized the basic ideas and methods of constructing and characterizing supramolecular oil field chemical materials, and introduced its applications in enhanced oil recovery, including cleaning fracturing fluid, improving oil displacement efficiency, assisting gas displacement, controlling CO_2_ migration rate, etc. Compared with conventional covalent polymers such as HPAM and xanthan gum, the supramolecular polymer introduced into non-covalent interactions is easier to adapt to the harsh reservoir environment, and it solves the contradiction between polymer injectivity and viscosity-increasing ability. At the same time, these polymers show various functions and are expected to replace many potential applications of HPAM in oil field chemistry fields such as drilling construction, hydraulic fracturing, enhanced oil recovery, oilfield wastewater treatment, profile control, etc. Supramolecular polymers have rheological properties common to conventional polymers, and at the same time, they make up for the defects of oil displacement materials in injectivity, temperature resistance, salt resistance and shear resistance. However, as a new research direction, there are still many challenges before mass production and field application. Due to the many problems caused by material replacement, the development direction of supramolecular polymers in this field is prospected from different perspectives.

Through the detailed introduction of the structure, oil displacement performance and EOR application of supramolecular polymers, we hope that this review can help some researchers engaged in oil and gas recovery to obtain a complete blueprint of the new research field of supramolecular polymers for EOR, and they have a new understanding of their molecular design, synthesis and characterization. Meanwhile, we believe that the design and synthesis of supramolecular polymers will lay a foundation for the research and development of new intelligent oilfield chemical materials as well as bring new opportunities and challenges to the oil field.

## Figures and Tables

**Figure 1 polymers-14-04405-f001:**
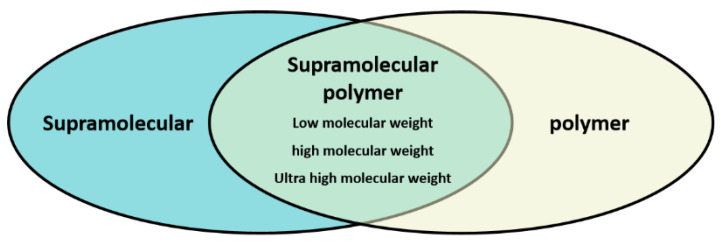
Category of supramolecular polymers.

**Figure 2 polymers-14-04405-f002:**
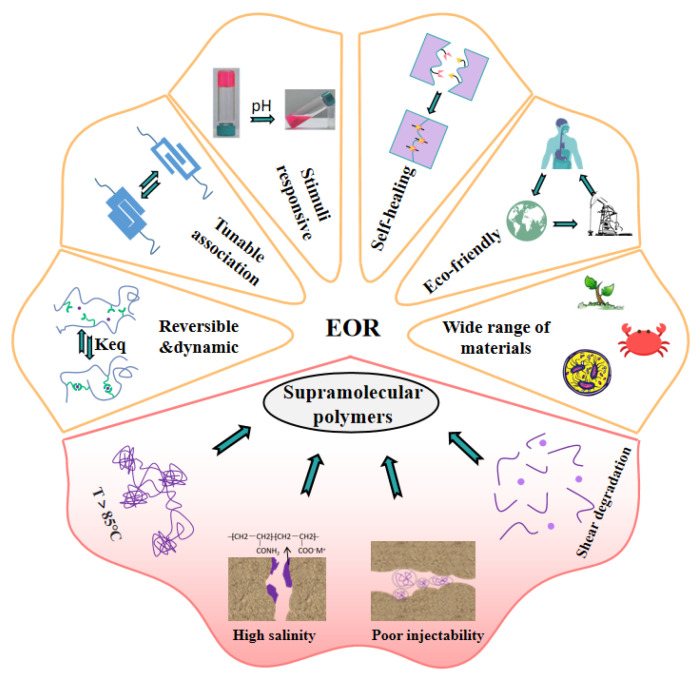
Display of supramolecular polymers and the limitations of conventional polymers. Supramolecular polymers are expected to solve the challenges faced by conventional oil displacement polymers with a series of unique advantages.

**Figure 3 polymers-14-04405-f003:**
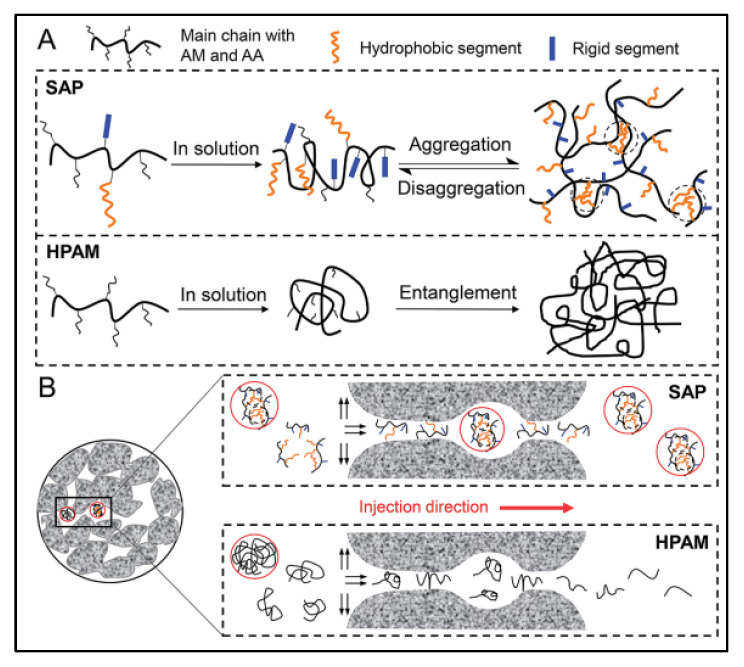
Comparison of the (**A**) morphologies and (**B**) flow behaviors of SAP and HPAM in porous media [73].

**Figure 4 polymers-14-04405-f004:**
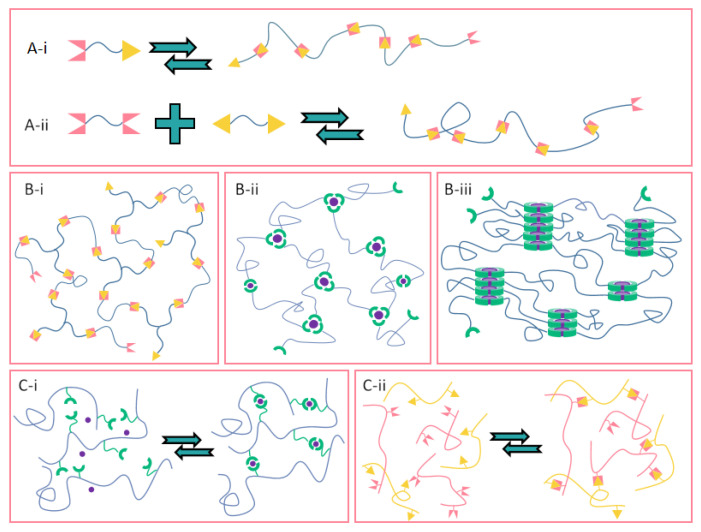
Different Design Strategies of Supramolecular Polymer Networks. (**A-i**) Linear supramolecules are linked by homogenous complementary motifs. (**A-ii**) Linear supramolecules are linked by heterocomplementary motifs. (**B-i**) Supramolecular networks are cross-linked in a bivalent manner through triplet components. (**B-ii**) Supramolecular networks are cross-linked by associative linear nodes with a functionality higher than two. (**B-iii**) Supramolecular networks are cross-linked by the lateral stacking or aggregation of dynamic cross-linking points. (**C-i**) Side chain supramolecular polymers are cross-linked by a low molecular weight cross-linker. (**C-ii**) Side-chain polymers chains cross-linked by mutual heterocomplementary polymer–polymer binding.

**Figure 5 polymers-14-04405-f005:**
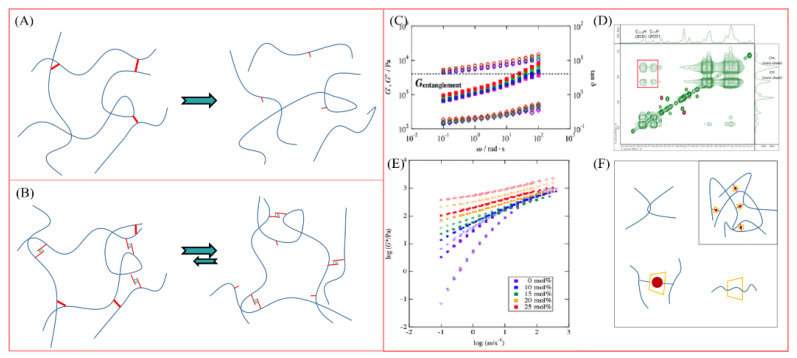
(**A**,**B**) Covalent polymer networks and supramolecular polymer networks (the red thick line represents covalent cross-linking, while the red thin line represents non-covalent cross-linking). (**C**) G′, G″, and tan δ of host–guest gels containing various concentrations of host molecules. The dashed line represents the plateau value of the entangled polyacrylamide solution with the same concentration [129]. (**D**) Two-dimensional (2D) NMR spectrum of the polyacrylamide solution containing β-CDs [25]. (**E**) Dynamic viscoelastic measurements of polyacrylamide containing only host molecules [25]. (**F**) Crosslinking mode of HPAM containing cyclodextrin adamantane complex.

**Figure 6 polymers-14-04405-f006:**
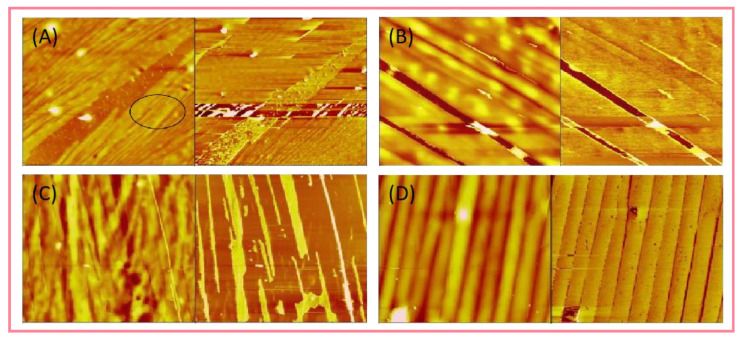
AFM topography and phase image of PPSA at different concentrations. (**A**) 0.5 g/L. (**B**) 0.7 g/L. (**C**) 1 g/L. (**D**) 1.5 g/L [132].

**Figure 7 polymers-14-04405-f007:**
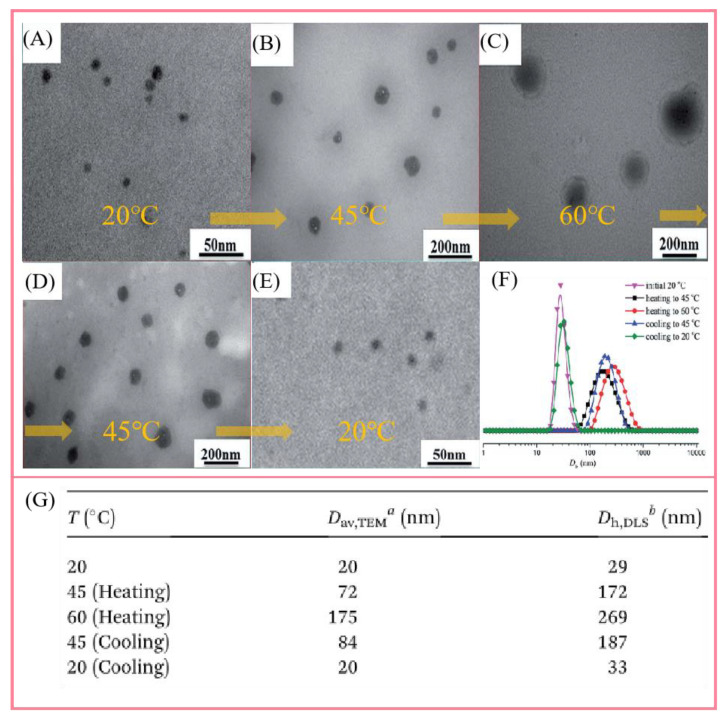
Typical TEM images of PNIPAM-(2CD-2MPEG) aqueous solutions (**A**–**E**) with a concentration of 1.0 mg·mL^−1^ during the heating–cooling process. (**F**–**G**) The DLS results show a self-assembly process similar to that of TEM [138].

**Figure 8 polymers-14-04405-f008:**
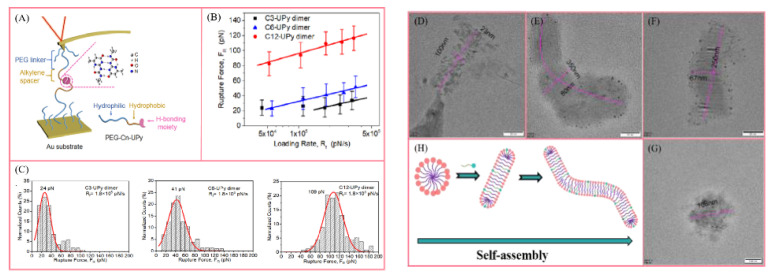
(**A**–**C**) The synergistic effect between hydrogen bonding and hydrophobic interaction was studied by Chen and coworkers [79]. (**D**–**H**) Through Cryo-TEM, our group revealed the solubilization mechanism of surfactant and determined the solubilization site [142].

**Figure 9 polymers-14-04405-f009:**
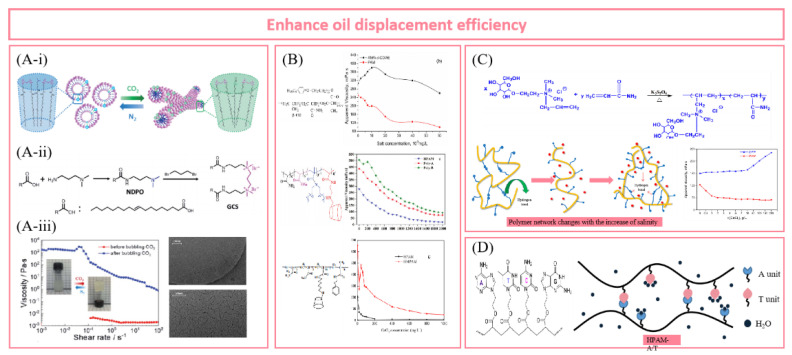
Display of the application of the supramolecular polymers in enhancing oil displacement efficiency. (**A**) A CO_2_-responsive smart system based on the charge transition of the hydrophilic head of a single-chain surfactant [148]. (**B**) The supramolecular polymer based on host–guest interaction has the phenomenon of salt thickening [35,37,95]. (**C**) Due to the coordination between hydrogen bond and metal, HPAM with a glucoside unit showed super high salt tolerance [149]. (**D**) Introducing bases into the side chain of HPAM [150].

**Figure 10 polymers-14-04405-f010:**
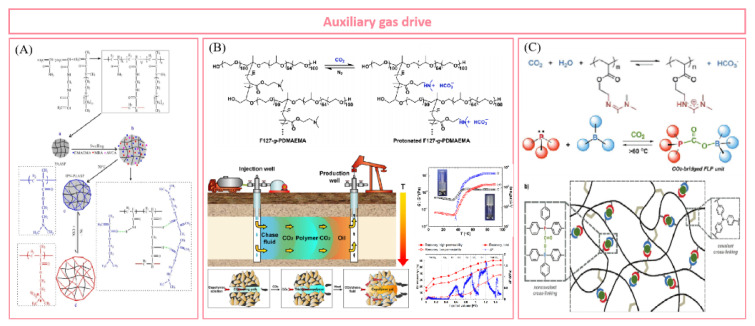
Display of the application of the supramolecular polymers in auxiliary gas drive. (**A**) CO_2_-responsive interpenetrating network based on covalent polymer and supramolecular polymer [155]. (**B**) CO_2_-responsive polymer assisted WAG flooding [156]. (**C**) The bridged FLP unit can be consid-ered as an attractive supramolecular synthon for the creation of supramolecular functional materials [157].

**Table 1 polymers-14-04405-t001:** Superiority of supramolecular polymers over conventional oil displacement polymers and the inevitable problems in their application in improving oil recovery.

	Conventional Polymer Materials	Supramolecular Polymer
Source and cost of materials	Limited, petroleum-based monomer Low cost	Wide range of source [160,161,162] Medium cost
Characterization items	Molecular structure Rheological properties	Self-assembly research [163] Binding constant
Polymer network	Static, permanent network [164,165,166] Chemical covalent cross-link	Reversible, dynamic network Both physical non-covalent and chemical covalent cross-link
Environment compatibility	Poor [167,168]	Not yet clear
Stimulus responsiveness	Poor	pH, temperature, CO_2_, pressure, ion concentration response
Thickening ability	The higher the molecular The greater the viscosity	Good
Temperature and salt tolerance	Degradation under high temperature and high salinity	Insensitive to temperature and salinity
Shear resistance	Low viscosity retention	High viscosity retention
Injectability	Difficult to inject Cause blockage Viscosity loss	Viscosity controllable Easy to inject
Applicable reservoir	Medium high-permeability Reservoir (50–10,000 mD) T < 85 °C,K < 60,000 mg·L^−1^	Applicable to many types of reservoirs

## Data Availability

Not applicable.

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
