# Peer review of "Can Supramolecular Polymers Become Another Material Choice for Polymer Flooding to Enhance Oil Recovery?"

_polymers, 2022, doi:10.3390/polym14204405_

Round 1

Reviewer 1 Report

This paper is a very detailed review on Polymer Flooding Materials
A section of abbreviations at the end of the text is necessary for an easy reading, especially since some acronyms are not always given in the text (e.g. page 5, HPAM (hydrolysed polyacrylamide), .....)
There is a problem with the title of section 2.2 which is the same as section 2.1
Line 257: give the ref. after 1978
Line 375: remove a ?
Line 453 and line 862: kJ (J is a capital teller)
Line 521: Figure 2 instead of 1
Figure 2A-i and 2A-ii: refer to it in the text
Line 531-543: put schema of Figure 6 and be consistent with the notations in Figure 6 (b-i instead of B-I in the text, .....)
Line 751: change viscosity modulus by loss and correct G' by G'', just after
Line 741- 761 add figures to highlight the mechanical properties of these systems
Lines 814-816 refer to Figure 4 and discuss it further in the text (heating-cooling)
Figures 5C-D: refer to these figures in the text
Figure 6: separate the figure into 3: 6- Hydraulic fracturing; 7-Enhance oil displacement efficiency; 8-Auxiliary gas drive, for clarity and be careful between the notations in the text and in Figures
section 6.2.3: It is necessary to add a discussion on the durability of these materials (long term behavior)

Author Response

Dear Dr Editors and reviewers,

Thank you for giving us the opportunity to submit a revised draft of the manuscript “Can supramolecular polymers become another material choice for polymerflooding to enhance oil recovery?” for publication in the Journal of polymers. We appreciate the time and effort that you and the reviewers dedicated to providing feedback on our manuscript and are grateful for the insightful comments on and valuable improvements to our paper.We have incorporated most of the suggestions made by the reviewers. Those changes are highlighted within the manuscript. Please see the attachment, in red, for a point-by-point response to the reviewers’comments and concerns. All page numbers refer to the revised manuscript file with tracked changes.  

We hope that all these changes fulfil the requirements to make the manuscript acceptable for publication in polymers.

Looking forward to hearing from you soon.

Sincerely yours,

Zhirong Zhang and Linhui Sun on behalf of the authors.

Corresponding author: Zhirong Zhang at Institute of Seepage Mechanics,University of Chinese Academy of Sciences, 065000, Langfang City, Hebei Province, China, [email protected],phone number: 18392055195

Reviewer 2 Report

The review manuscript deals with the application of supramolecular polymers for enhanced oil recovery application. Design strategies, interactions of such polymers and rheological properties were the key focus of the review. The manuscript needs major revision before its acceptance.

 The manuscript needs revision to fix the grammatical issues. For example “ to Can supramolecular polymers become another material choice for polymerflooding to enhance oil recovery?”

Some statements need rephrasing, for example “With the increase of oil consumption and the gradual decrease of crude oil production, oil workers are forced to develop new EOR technologies to extract crude oil from mature oilfields and low-grade reservoirs”.

Some sections are confusing. For example, the title of section 2.1 and 2.2 are similar “Conventional polymers utilized for EOR.”

A section should be added to differentiate between the high molecular weight and supramolecular weight polymers.

I believe the authors are not clear about the term EOR. In section 5, the authors are trying to make hydraulic fracturing a type of EOR operation. Hydraulic fracturing is not an EOR operation. I suggest the authors change the title from EOR to oilfield applications.

I did not see a single structure of these supramolecular polymers. I suggest authors to report the available structure of such polymers

Thermal stability is a key parameter of oilfield applications. I suggest authors add a paragraph on this. 

Author Response

Dear Dr Editors and reviewers,

Thank you for considering the revised version of our manuscript(polymers-1940536) “Can supramolecular polymers become another material choice for polymerflooding to enhance oil recovery?”, by Linhui Sun. et al. for publication in polymers. We are thankful to the referees and the Editor for pointing out some important modifications needed in the report. We have thoughtfully taken into account these comments. The explanation of what we have changed in response to the reviewers’ concerns is given point by point in the attachment.We believe that the comments have been highly constructive and very useful to restructure the manuscript.  

We hope that all these changes fulfil the requirements to make the manuscript acceptable for publication in polymers.

Looking forward to hearing from you soon.

Sincerely yours,

Zhirong Zhang and Linhui Sun on behalf of the authors.

Corresponding author: Zhirong Zhang at Institute of Seepage Mechanics,University of Chinese Academy of Sciences, 065000, Langfang City, Hebei Province, China, [email protected],phone number: 18392055195

Round 2

Reviewer 2 Report

The revised manuscript can be accepted in its current form as the authors have addressed majority of the comments raised by the reviewer.